# Can we make sense out of "Tensor Field Theory"?

**Vincent Rivasseau**[1†] **and Fabien Vignes-Tourneret**[2⋆]

**1** Laboratoire de physique des 2 infinis Irène Joliot-Curie,
CNRS, Université Paris-Saclay, 91405 Orsay Cedex, France.
**2** Univ Lyon, CNRS, Université Claude Bernard Lyon 1,
UMR 5208, Institut Camille Jordan, F-69622 Villeurbanne, France.

† rivass@th.u-psud.fr, ⋆ vignes@math.univ-lyon1.fr

## Abstract

We continue the constructive program about tensor field theory through the next natural model, namely the rank five tensor theory with quartic melonic interactions and propagator inverse of the Laplacian on $U(1)^5$. We make a first step towards its construction by establishing its power counting, identifying the divergent graphs and performing a careful study of (a slight modification of) its RG flow. Thus we give strong evidence that this just renormalizable tensor field theory is non perturbatively asymptotically free.

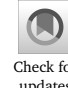
# 1   Introduction

Recently Hairer [1] solved a series of stochastic differential equations such as the KPZ equation or the $\phi_3^4$ equation. An advantage of such equations is that they are better suited to Monte Carlo computations than functional integrals. Since then, in a systematic series of impressive articles, Hairer and his collaborators [2–4] extended their initial programme to cover the BPHZ renormalization [5–7]. In contrast to dimensional renormalization, BPHZ renormalization is adapted to the program of constructive field theory. It incorporates the multiscale expansion, a main constructive tool [8], and a more up-to-date mathematical formulation of renormalization based on Hopf algebras [9].

To the attentive observer, constructive field theory, namely the point of view which Hairer called *static*, is rapidly merging into the *regularity structures and corresponding models* of Hairer, which he called the *dynamic* point of view. In the language of quantum field theory, it happens that the equations which Hairer solved were all Bosonic super-renormalizable. Now is time for advancing the next step: the Bosonic just-renomalizable quantum field theories. The BPHZ renormalization was initially designed to cover theories such as QED in dimension four, the main theory at the time. But a profound objection were raised, initially by Landau. Now we have a name for that obstacle: QED is *not asymptotically free*. Fortunately for the future of quantum field theory, the discovery that electroweak and strong interactions are asymptotically free were instrumental in its "rehabilitation" as a fundamental theory.

A famous theorem due to Coleman states that any *local Bosonic asymptotically free field theory must include non-Abelian gauge theories*. Non-Abelian gauge theories lead to an additional severe problem: the presence of Gribov ambiguities [10] due to gauge fixing. The way out of these difficulties is a main reason for considering the stochastic quantization [11], since in this method there are no need to fix the gauge, so no need to solve Gribov ambiguities. But it remains still a tough programme.

On the road to this lofty goal, we propose an intermediate step which might be worth the effort in itself. It *escapes* Coleman's theorem by being a *non-local* theory. We have in mind the tensor field theory. Born in the quantum gravity craddle [12–14], random tensor models extend random matrix models and therefore were introduced as promising candidates for an *ab initio* quantization of gravity in rank/dimension higher than 2. However their study is less advanced since they lacked for a long time an analog of the famous 't Hooft $1/N$ expansion for random matrix models. Their modern reformulation [15–17] considers *unsymmetrized*

random tensors,[1] a crucial improvement which let the large $N$ limit appear [18–20]. The limit of large matrix models is made of planar graphs. Surprisingly perhaps, the key to the $1/N$ tensors is made of a new and *simpler* class of Feynman graphs that we called *melonic*. They form the dominant graphs in this limit [21, 22].[2]

Random tensor models can be further divided into fully invariant models, in which both propagator and interaction are left invariant by the symmetry (such as $U(N)^{\otimes d}$), and *non-local field theories* where the propagator is for example the ordinary Laplacien on the torus $U(1)^{\otimes d}$ (which breaks the symmetry) but in which the interaction is left invariant by the symmetry. To our own surprise, such just-renormalizable models turn out to be asymptotically free [23, 24]. In particular the simplest such model in this category, nicknamed $T_5^4$ theory is asymptotically free! It made them an ideal playground for advancing the mathematics both in the static sense of constructive theory and in the sense of Hairer's stochastic quantization. This fact now many years old was perhaps overlooked by the theoretical and mathematical physics community.

Also the tensor methods and models in quantum gravity that one of us baptized the tensor track [25–30] was given a big boost from an unexpected corner. Since the advent of the SYK model [31–33] it appears that 1-dimensional quantum random tensor is even richer than the 0-dimensional ordinary random tensor theory [34–37]. It is approximately reparametrization invariant (*i.e.* conformal), includes holography and it saturates the MSS bound [38].

In fact the real applications, as it often happens, might be elsewhere. Today we probe reality by multiples sensors. That is, we represent that reality by multidimensional big arrays which are, in the mathematical sense, nothing but *big tensors*. Hence we need to develop better and more versatile algorithms to probe tensors in this limit. Such algorithms could benefit of the modern formulation of random tensors. This is especially true for those separating signal to noise. One example is tensor PCA [39–42], which extends classical matrix PCA to tensors. Such algorithms could be applied in a variety of domains, high energy physics (detection of particule trajectories), spectral imaging or videos, neuroimaging, chemometrics, pharmaceutics, biometrics, social networks and many more. In fact the analysis of big tensors form a bottleneck in such a dazzling kaleidoscope that it is no exaggeration to say that any main progress in this field may create a revolution in artificial intelligence.

Now let us come down to earth. The *tensor theory new constructive program* [43] is well advanced in the super-renormalizable case [44, 45]. In [46] the $U(1)$ rank-three model with inverse Laplacian propagator and quartic melonic interactions, which we nickname $T_3^4$, was solved. In [47] the $U(1)$ rank-four model $T_4^4$ was solved. This model looks comparable in renormalization difficulty to the ordinary $\phi_3^4$ theory, but non-locality and the graphs are more complex hence requires several additional non-trivial arguments. The next goal is to treat just-renormalizable asymptotically free Bosonic $T_5^4$. In 1979, G. 't Hooft gave a series of lectures entitled *Can we make sense out of "Quantum Chromodynamics"?* [48].[3] He presented there arguments and strategies to control QCD via the study of its singularities in the Borel plane. To this aim, he had to control the flow of the coupling constant in the complex plane. The tensor field theory $T_5^4$ is a perfect playground for constructive purposes as its flows can be controled precisely thanks to its simple and exponentially bounded divergent sector. In the present paper we make a further step by connecting it, modulo certain hypotheses, to an autonomous non-linear flow of the theory of dynamical systems.

The $T_5^4$ tensor field theory is precisely defined in Section 2. In particular, we present the cut-offs we use and an alternative representation of the model in terms of an intermediate

---

[1]*Symmetrized* random tensors are more difficult but melons still dominate again [49, 50], at least in rank 3.

[2]In quantum field theory an early reference at those Feymnam graphs appear in [51]

[3]The title of our article refers obviously to this seminal work.

matrix field. Section 3 is devoted to the three different representations of Feynman graphs we need (tensor graphs, coloured graphs and intermediate field maps) as well as related concepts thereof. In Section 4 we derive the power-counting, identify the families of divergent graphs and give the recursive definitions of the *melonic* correlation functions. For constructive purposes, we will employ none of the bare, renormalized or even fully effective perturbative expansions. In fact, it will be preferable to fully mass renormalize the correlation functions but use effective wave-functions and coupling constants. We define all these objects in Section 5. We also prove there that effective wave-functions and coupling constants are analytic functions of the bare coupling. The main result of Section 5 is Theorem 5.1 which consists in a non perturbative definition of the RG flow for the coupling constant. A careful study of an approximation of this flow is carried out in Section 6 using tools and concepts from discrete and continuous holomorphic local dynamical systems. We identify in particular "cardioid-like" domains of the complex plane invariant under this modified RG flow, see Theorems 6.1 and 6.3 and Corollary 6.4.

Solving this $T_5^4$ model means defining its correlation functions non perturbatively in the coupling constant $g$. More precisely it requires to prove the existence of holomorphic functions of $g$ in a (probably cardioid-like, with a cut on the negative real axis) domain of the complex $g$-plane such that their Taylor expansions coincide with the perturbative expansions of the (formal) correlation functions of the model. Moreover these functions should very probably be proven Borel summable.

To achieve that goal, one expresses the regularized and renormalized correlation functions as series of analytic functions, normally convergent in a domain the size of which is uniformly bounded in the ultraviolet cutoff. The infinite cutoff limit is then well-defined and analytic. These expansions consist in partial resummations of the perturbative series, either expressed in an intermediate field representation (this is the so-called Loop Vertex Expansion [46, 47, 52]) or obtained from a specific change of the initial tensor field variables (in which case it is called Loop Vertex Representation [53–55]). Both approaches have pros and cons but none of them appears totally suited for the new challenges brought by the $T_5^4$ theory. An update of all currently known approaches to constructive tensor field theory seems necessary [56].

## 2 The model

### 2.1 Rank 5 tensors and free Gaussian measure

In this section we follow as closely as possible the notations of [47]. Consider a pair of conjugate rank-5 tensor fields

$$T_{\boldsymbol{n}}, \overline{T}_{\overline{\boldsymbol{n}}}, \text{ with } \boldsymbol{n} = (n_1, n_2, n_3, n_4, n_5) \in \mathbb{Z}^5, \overline{\boldsymbol{n}} = (\overline{n}_1, \overline{n}_2, \overline{n}_3, \overline{n}_4, \overline{n}_5) \in \mathbb{Z}^5.$$

They belong respectively to the tensor product $\mathcal{H}^{\otimes} := \mathcal{H}_1 \otimes \mathcal{H}_2 \otimes \mathcal{H}_3 \otimes \mathcal{H}_4 \otimes \mathcal{H}_5$ and to its dual, where each $\mathcal{H}_i$ is an independent copy of $\ell_2(\mathbb{Z}) = L_2(U(1))$, and the colour or strand index $i$ takes values in $\{1, 2, 3, 4, 5\}$. By Fourier transform, the field $T$ can be considered also as an ordinary scalar field $T(\theta_1, \theta_2, \theta_3, \theta_4, \theta_5)$ on the five-dimensional torus $\mathbb{T}_5 = U(1)^5$ and $\overline{T}(\overline{\theta}_1, \overline{\theta}_2, \overline{\theta}_3, \overline{\theta}_4, \overline{\theta}_5)$ is simply its complex conjugate [46, 57]. The tensor index $\boldsymbol{n}$ can be thought as the *momenta* associated to the positions $\boldsymbol{\theta}$.

Throughout this paper, we always use bold characters to denote tuples of at least two variables.

We introduce the normalized Gaussian measure

$$d\mu_C(T,\overline{T}) := \left( \prod_{\boldsymbol{n},\overline{\boldsymbol{n}}} \frac{dT_{\boldsymbol{n}} d\overline{T}_{\overline{\boldsymbol{n}}}}{2i\pi} \right) \det(C^{-1}) \, e^{-\sum_{\boldsymbol{n},\overline{\boldsymbol{n}}} T_{\boldsymbol{n}} C^{-1}_{\boldsymbol{n},\overline{\boldsymbol{n}}} \overline{T}_{\overline{\boldsymbol{n}}}},$$

where the covariance $C$ is

$$C_{\boldsymbol{n},\overline{\boldsymbol{n}}} = \delta_{\boldsymbol{n},\overline{\boldsymbol{n}}} C(\boldsymbol{n}), \qquad C(\boldsymbol{n}) = \frac{1}{\boldsymbol{n}^2 + m^2}, \qquad \boldsymbol{n}^2 := n_1^2 + n_2^2 + n_3^2 + n_4^2 + n_5^2. \tag{2.1}$$

This defines the *free* tensor fields as random distributions on $\mathbb{Z}^5$, namely on the dual of rapidly decreasing sequences on $\mathbb{Z}^5$. But as we are interested in interacting tensor fields, we need to regularise the free measure.

## 2.2 Ultraviolet cutoff

In practice we want to restrict the index $\boldsymbol{n}$ to lie in a finite set rather than $\mathbb{Z}^5$ in order to have a well-defined proper (finite dimensional) tensor model. This restriction is an ultraviolet cutoff in quantum field theory language.

A colour-factorized ultraviolet cutoff would restrict all previous sums over tensor indices to lie in $[-N,N]$. However it is not well adapted to the rotation invariant propagator of eq. (2.1) below, nor very convenient for multi-slice analysis as in [58]. Therefore we introduce a rotation invariant cutoff but in contrast with [47] it will be smooth.

Let $a,\epsilon$ be two positive numbers such that $\epsilon < a$. Let $\chi_\epsilon$ be a smooth positive function with support $[-\epsilon,\epsilon]$. We denote by $\mathbf{1}_{[-a,a]}$ the indicator function of $[-a,a]$. In order to prepare for multiscale analysis (see Section 5.3), we fix an integer $M > 1$ (as ratio of a geometric progression $M^j$) and choose a large integer $j_{\max}$. Our ultraviolet cutoff is defined as

$$\kappa_{j_{\max}}(\boldsymbol{n}^2) := \kappa(M^{-2j_{\max}} \boldsymbol{n}^2), \quad \kappa(\boldsymbol{n}^2) := \mathbf{1}_{[-a,a]} \star \chi_\epsilon(\boldsymbol{n}^2).$$

It is smooth, positive, compactly supported, and satisfies

$$\kappa_{j_{\max}}(\boldsymbol{n}^2) = \begin{cases} 0 & \text{if } \boldsymbol{n}^2 > (a+\epsilon)M^{2j_{\max}}, \\ 1 & \text{if } 0 \leqslant \boldsymbol{n}^2 \leqslant (a-\epsilon)M^{2j_{\max}}. \end{cases}$$

It is convenient to choose $a = 5/2$ and $\epsilon = 3/2$ so that the UV cutoff $\kappa$ effectively restricts each colour index to lie in $[-N,N]$ with

$$N := \lfloor (a+\epsilon)^{1/2} M^{j_{\max}} \rfloor = 2M^{j_{\max}}.$$

The normalized bare Gaussian measure with cutoff $j_{\max}$ is

$$d\mu_{C_b}(T,\overline{T}) := \left( \prod_{\boldsymbol{n},\overline{\boldsymbol{n}}} \frac{dT_{\boldsymbol{n}} d\overline{T}_{\overline{\boldsymbol{n}}}}{2i\pi} \right) \det(C_b^{-1}) \, e^{-\sum_{\boldsymbol{n},\overline{\boldsymbol{n}}} T_{\boldsymbol{n}} C^{-1}_{b;\boldsymbol{n},\overline{\boldsymbol{n}}} \overline{T}_{\overline{\boldsymbol{n}}}},$$

where the bare covariance $C_b$ is, up to a bare field strength parameter $Z_b$, the inverse of the Laplacian on $\mathbb{T}_5$ with momentum cutoff $j_{\max}$ plus a bare mass term

$$C_{b;\boldsymbol{n},\overline{\boldsymbol{n}}} = \delta_{\boldsymbol{n}\overline{\boldsymbol{n}}} \kappa_{j_{\max}}(\boldsymbol{n}^2) C_b(\boldsymbol{n}), \qquad C_b(\boldsymbol{n}) = \frac{1}{Z_b} \frac{1}{\boldsymbol{n}^2 + m_b^2}, \qquad \boldsymbol{n}^2 := n_1^2 + n_2^2 + n_3^2 + n_4^2 + n_5^2.$$

A random tensor $T$ distributed according to the measure $\mu_{C_b}$ is almost surely a smooth function on $U(1)^5$.

## 2.3 The bare model

The generating function for the moments of the model is

$$\mathcal{Z}_b^{(N)}(g_b, J, \overline{J}) = \mathcal{N}^{-1} \int e^{T \cdot \overline{J} + J \cdot \overline{T}} e^{-\frac{g_b Z_b^2}{2} \sum_c V_c(T, \overline{T})} d\mu_{C_b}(T, \overline{T}), \tag{2.2}$$

where the scalar product of two tensors $A \cdot B$ means $\sum_n A_n B_{\underline{n}}$, $g_b$ is the *bare* coupling constant (which depends on the cutoff $N$), the source tensors $J$ and $\overline{J}$ are dual respectively to $\overline{T}$ and $T$ and $\mathcal{N}$ is a normalization factor. To compute correlation functions it is common to choose

$$\mathcal{N} = \int \exp\left(-\frac{g_b Z_b^2}{2} \sum_c V_c(T, \overline{T})\right) d\mu_{C_b}(T, \overline{T}),$$

which is the sum of all vacuum bare amplitudes. However following the constructive tradition, we shall limit $\mathcal{N}$ to be the exponential of the (infinite) sum of the *divergent* connected vacuum amplitudes. Remark the $Z_b^2$ scaling factor multiplying $g_b$ in eq. (2.2).

To make the interaction $\sum_c V_c(T, \overline{T})$ in eq. (2.2) explicit, we recall first some notation. Tr, $\mathbb{I}$ and $\langle , \rangle$ mean respectively the trace, the identity and the scalar product on $\mathcal{H}^{\otimes}$. $\mathbb{I}_c$ is the identity on $\mathcal{H}_c$, $\mathrm{Tr}_c$ is the trace on $\mathcal{H}_c$ and $\langle , \rangle_c$ the scalar product restricted to $\mathcal{H}_c$. The notation $\hat{c}$ means "every colour except $c$". For instance, $\mathcal{H}_{\hat{c}}$ means $\bigotimes_{c' \neq c} \mathcal{H}_{c'}$, $\mathbb{I}_{\hat{c}}$ is the identity on the tensor product $\mathcal{H}_{\hat{c}}$, $\mathrm{Tr}_{\hat{c}}$ is the partial trace over $\mathcal{H}_{\hat{c}}$ and $\langle , \rangle_{\hat{c}}$ the scalar product restricted to $\mathcal{H}_{\hat{c}}$.

$T$ and $\overline{T}$ can be considered both as vectors in $\mathcal{H}^{\otimes}$ or as diagonal (in the momentum basis) operators acting on $\mathcal{H}^{\otimes}$, with eigenvalues $T_n$ and $\overline{T}_{\overline{n}}$. An important quantity in melonic tensor models is the partial trace $\mathrm{Tr}_{\hat{c}}[T\overline{T}]$, which we can also identify with the partial product $\langle T, \overline{T} \rangle_{\hat{c}}$. It is a (in general non-diagonal) operator in $\mathcal{H}_c$ with matrix elements in the momentum basis

$$\langle T, \overline{T} \rangle_{\hat{c}}(n_c, \overline{n}_c) = \mathrm{Tr}_{\hat{c}}[T\overline{T}](n_c, \overline{n}_c) = \left[\prod_{c' \neq c}\left(\sum_{n_{c'}, \overline{n}_{c'}} \delta_{n_{c'} \overline{n}_{c'}}\right)\right] T_n \overline{T}_{\overline{n}}.$$

The main new feature of tensor models compared to ordinary field theories is the non-local form of their interaction, which is chosen invariant under independent unitary transformations on each colour index. In this paper we consider only the quartic melonic interaction [44], which is a sum over colours $\sum_{c=1}^5 V_c(T, \overline{T})$ where

$$\begin{aligned} V_c(T, \overline{T}) &= \mathrm{Tr}_c\left[(\mathrm{Tr}_{\hat{c}}[T\overline{T}])^2\right] \\ &= \sum_{\substack{n_c, \overline{n}_c, \\ m_c, \overline{m}_c}} \left(\sum_{n_{\hat{c}}, \overline{n}_{\hat{c}}} T_n \overline{T}_{\overline{n}} \delta_{n_{\hat{c}} \overline{n}_{\hat{c}}}\right) \delta_{n_c \overline{m}_c} \delta_{m_c \overline{n}_c} \left(\sum_{m_{\hat{c}}, \overline{m}_{\hat{c}}} T_m \overline{T}_{\overline{m}} \delta_{m_{\hat{c}} \overline{m}_{\hat{c}}}\right). \end{aligned}$$

This model is globally symmetric under colour permutations. It is just renormalizable like ordinary $\phi_4^4$ but unlike ordinary $\phi_4^4$ it is asymptotically free and using this crucial difference, we aim, in a future work, at making rigorous sense of it.

Mainly in order to prepare for the constructive study of the $T_5^4$ model, we present here its **intermediate field representation** [45]. We put $g_b =: \lambda_b^2$ and decompose the five interactions $V_c$ in eq. (2.2) by introducing five intermediate Hermitian $N \times N$ matrix[4] fields $\sigma_c^t$ acting on $\mathcal{H}_c$ (here the superscript $t$ refers to transposition) and dual to $\mathrm{Tr}_{\hat{c}}[T\overline{T}]$, in the following way

$$e^{-\frac{(\lambda_b Z_b)^2}{2} V_c(T, \overline{T})} = \int e^{i\lambda_b Z_b \mathrm{Tr}_c\left[\left(\mathrm{Tr}_{\hat{c}}[T\overline{T}]\right)\sigma_c^t\right]} d\nu(\sigma_c^t),$$

---

[4]The indices of $\sigma$ cannot be bigger than the maximal value $N$ of each tensor index.

where $d\nu$ is the usual GUE measure, that is $d\nu(\sigma_c^t) = d\nu(\sigma_c)$ is the normalized Gaussian independently identically distributed measure of covariance 1 on each coefficient of the Hermitian matrix $\sigma_c$. It is convenient to consider $C_b$ as a (diagonal) operator acting on $\mathcal{H}^\otimes$, and to define in this space the operator

$$\sigma := \sum_c \sigma_c \otimes \mathbb{I}_{\hat{c}} = \sigma_1 \otimes \mathbb{I}_2 \otimes \mathbb{I}_3 \otimes \mathbb{I}_4 \otimes \mathbb{I}_5 + \cdots + \mathbb{I}_1 \otimes \mathbb{I}_2 \otimes \mathbb{I}_3 \otimes \mathbb{I}_4 \otimes \sigma_5.$$

Performing the now Gaussian integration over $T$ and $\overline{T}$ yields

$$\mathcal{Z}_b^{(\mathrm{N})}(g, J, \overline{J}) = \mathcal{N}^{-1} \iint e^{T \cdot \overline{J} + J \cdot \overline{T}} e^{i\lambda_b Z_b \operatorname{Tr}_c\left[\left(\operatorname{Tr}_{\hat{c}}\left[T\overline{T}\right]\right)\sigma_c^t\right]} d\mu_{C_b}(T, \overline{T}) \prod_c d\nu(\sigma_c)$$

$$= \mathcal{N}^{-1} \int e^{\langle \overline{J}, R(\sigma) C_b J \rangle - \operatorname{Tr} \log(\mathbb{I} - i\lambda_b Z_b C_b \sigma)} d\nu(\sigma), \tag{2.3}$$

where $d\nu(\sigma) := \prod_c d\nu(\sigma^c)$, and $R$ is the **resolvent** operator on $\mathcal{H}^\otimes$

$$R(\sigma) := \frac{1}{\mathbb{I} - i\lambda_b Z_b C_b \sigma}.$$

# 3 Feynman graphs

Perturbative expansions in quantum field theory are indexed by graphs called Feynman graphs. Their properties reflect analytical aspects of the action functional. Here we will deal with three different graphical notions.

## 3.1 Tensor graphs

The first one corresponds to the Feynman graphs of action (2.2) in which the fields are tensors of rank five. As for random matrix models, Feynman graphs are stranded graphs (so-called ribbon graphs in the matrix case) where each strand represents the conservation of one tensor index. The corresponding Feynman rules are recalled in Fig. 1 where an example of such a Feynman graph is also given. Such graphs will be called **tensor graphs** in the sequel and denoted by emphasized letters such as $\mathbb{G}$. A tensor graph is open if it has a positive number of external edges[5] and closed otherwise. An open graph with $n$ external edges is often called an **$n$-point graph**.

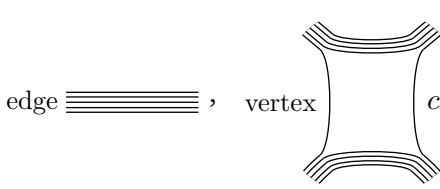
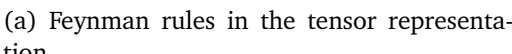
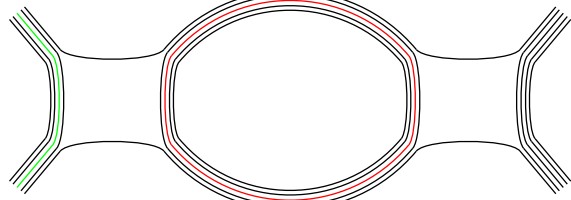

(a) Feynman rules in the tensor representation.

(b) An open tensor graph with 2 vertices, 2 internal and 4 external edges. One face is drawn in red, one external path in green.

Figure 1: Tensor graphs.

---

[5]What we call external edges are actually half-edges and open graphs are in fact pre-graphs. But we do not insist on being so precise with our terminology.

The power-counting, *i.e.* the behaviour at large $N$ of the amplitude, of a tensor graph $\mathbb{G}$ depends on the number $F(\mathbb{G})$ of its cycles, also called **faces**. Open tensor graphs also have non cyclic strands which we call **external paths**, see Fig. 1b. It will be convenient to express the number of faces in terms of the (reduced) Gurau degree [59] of the **coloured extension** of $\mathbb{G}$. We now explain these notions.

## 3.2 Coloured graphs

Strands of a tensor graph correspond to indices of the original tensor fields $T$ and $\overline{T}$. Each such index is labelled by an integer from 1 to 5 recalling that $T$ is an element of $\mathcal{H}_1 \otimes \mathcal{H}_2 \otimes \cdots \otimes \mathcal{H}_5$. We can then associate bijectively to any tensor graph $\mathbb{G}$ a bipartite 6-regular properly edge-coloured graph $G$ called its coloured extension. See Fig. 2 for a pictorial explanation of the bijection as well as an example. Such edge-coloured graphs, with or without the constraint of being 6-regular, will be called **coloured graphs** for simplicity and their symbols will be written in normal font. A $(D+1)$-regular coloured graph will simply be called $(D+1)$-coloured graph.

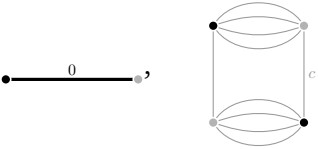

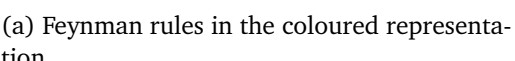

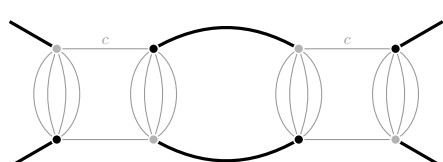

(a) Feynman rules in the coloured representation.

(b) The 6-coloured graph which is in bijection with the tensor graph of Fig. 1b.

Figure 2: Coloured graphs.

We will need several different notions associated to coloured graphs. The coloured extension of a closed (resp. open) tensor graph will also be considered closed (resp. open). In this work, edges of coloured graphs will bear a "colour" in $[5] := \{0, 1, \ldots, 5\}$. We will also write $[5]^*$ for the set $\{1, 2, \ldots, 5\}$. Let $G$ be a coloured graph. We let **col**$(G)$ be the set of colours labelling at least one edge of $G$. Let $c$ be an element of $[5]$. We will often write $\hat{c}$ for $[5] \setminus \{c\}$. We denote by $E_c(G)$ the set of edges of $G$ of colour $c$. The elements of $E_c(G)$ are the $c$-edges of $G$. If $\mathcal{C}$ is a subset of $[5]$, we denote $\bigcup_{c \in \mathcal{C}} E_c(G)$ by $E_{\mathcal{C}}(G)$. Let $E'$ be any subset of the edges of $G$, we let $G[E']$ be the spanning subgraph of $G$ induced by the edges in $E'$: the vertex-set of $G[E']$ is the same as the one of $G$, and the edge-set of $G[E']$ is $E'$.

Certain (coloured) subgraphs of coloured graphs play a particularly important role. We let again $\mathcal{C}$ be a subset of $[5]$. A $\mathcal{C}$-**bubble** is a connected component of $G[E_{\mathcal{C}}]$. Let $n$ be an element of $\{0, 1, \ldots, 6\}$. An $n$-bubble $B$ is a bubble such that col$(B)$ has cardinality $n$. A 2-bubble of a *closed* coloured graph is therefore a cycle whose edges bear two alternating colours. Cyclic 2-bubbles of $G$ whose colour set belong to $\{\{0, i\}, i \in [5]^*\}$ correspond to the faces of the corresponding tensor graph $\mathbb{G}$. By extension, cyclic 2-bubbles are often also called faces and their number denoted $F(G)$. The number of $\{0, c\}$-bubbles will be written $F_{0c}$ and we define $F_0(G) := \sum_{c \in [5]^*} F_{0c}$ so that $F_0(G) = F(\mathbb{G})$. Similarly, we denote by $F_{\emptyset}(G)$ the total number of faces of $G$, both colours of which are different from 0. Non cyclic 2-bubbles of $G$ represent the external paths of $\mathbb{G}$. The interaction vertices of a tensor graph $\mathbb{G}$ are in bijection with the $\hat{0}$-bubbles of its coloured extension $G$.

The (reduced) **Gurau degree** $\delta(G)$ of a *closed* $(D+1)$-coloured graph[6] $G$ is defined as

---

[6]In this case, by convention, the set of colours of $G$ is $[D]$.

follows [59]:

$$\delta(G) := \tfrac{1}{4}D(D-1)V(G) + D\,C(G) - F(G),$$

where $V(G)$ is the number of vertices of $G$ and $C(G)$ its number of connected components. It is a non negative integer. One can indeed show that it is the sum of the genera of some maps associated to $G$ [20]. Let $\mathfrak{S}^{(c)}_{[D]}$ be the set of cyclic permutations of $[D]$ and $\tau$ be such a permutation. Let $\mathfrak{J}_\tau(G)$ ($\mathfrak{J}_\tau$ if the context is clear) be the map whose underlying graph is $G$ and whose cyclic ordering of the edges around vertices is given by $\tau$. Such maps are called **jackets** in the tensor field literature [60], see Fig. 3 for an example. Then, we have

$$\delta(G) = \frac{1}{(D-1)!} \sum_{\tau \in \mathfrak{S}^{(c)}_{[D]}} g_{\mathfrak{J}_\tau}.$$

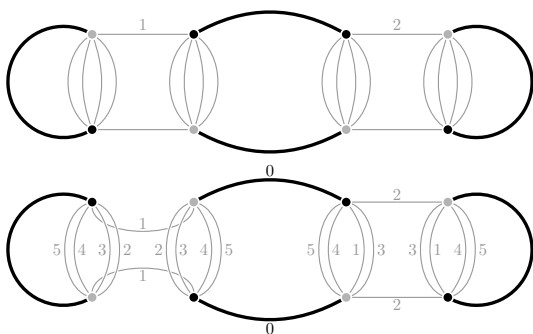

Figure 3: The jacket $\mathfrak{J}_{(023145)}$ (bottom) of the top coloured graph.

In order to classify the divergent graphs of tensor models, one needs an extension of the Gurau degree to *open* coloured graphs and the notion of boundary graph of a coloured graph. To start with, we need a slightly generalised version of jacket.[7]

**Definition 3.1 (Jacket of a possibly open coloured graph)** *Let $G$ be a $(D+1)$-coloured graph, open or closed. Let $\tau$ be a cyclic permutation of $[D]$. The jacket $\mathfrak{J}_\tau$ of $G$, with respect to $\tau$, is the map built in the following way:*

1. *consider $G$ as a graph,*

2. *fix the cyclic ordering of its edges around its vertices according to $\tau$,*

3. *delete the half- (or external) edges of this map.*

Before defining a natural version of the Gurau degree for possibly open coloured graphs, we need to remind the reader of the notion of a **boundary graph**. It is well-known that open (resp. closed) $(D+1)$-coloured graphs encode (triangulated) $D$-dimensional piecewise linear normal pseudo-manifolds with (resp. without) boundary [61, 62]. Let $G$ be such an open coloured graph. The boundary of its dual pseudo-manifold is triangulated by a complex dual to the boundary graph $\partial G$ of $G$. $\partial G$ is a $D$-coloured graph defined as follows: its vertex-set is the set of external edges of $G$. Its edge-set is the set of non cyclic 2-bubbles of $G$. Fig. 4 provides examples of boundary graphs. From the bijection between coloured graphs and tensor graphs, $\partial G$ defines the boundary (tensor) graph $\partial \mathbb{G}$ of $\mathbb{G}$.

---

[7]Such a map is called a pinched jacket in [57].

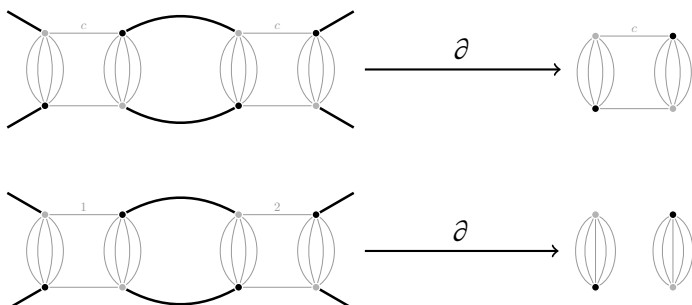

Figure 4: Boundary graphs.

**Definition 3.2 (Gurau degree of a possibly open coloured graph)** *The Gurau degree of a possibly open $(D + 1)$-coloured graph is defined as*

$$\delta(G) := \frac{1}{(D-1)!}\Big( \sum_{\tau \in \mathfrak{S}_{[D]}^{(c)}} g_{\mathfrak{J}_\tau(G)} - \sum_{\tau \in \mathfrak{S}_{[D]^*}^{(c)}} g_{\mathfrak{J}_\tau(\partial G)} \Big).$$

If $G$ is closed, $\partial G$ is empty and this equation reduces to the usual Gurau degree for closed graphs. This definition essentially originates from [57]. The point of interest for us is that, in the case where all the external edges of $G$ bear the same colour, it can be shown that $\delta(G)$ is a measure of the number of faces of $G$:

**Lemma 3.1** *Let G be a possibly open $(D+1)$-coloured graph. Its Gurau degree, given by Definition 3.2, follows*

$$\delta(G) = \tfrac{1}{4}D(D-1)V(G) + D\,C(G) - F(G) - C(\partial G) - \tfrac{1}{2}(D-1)E(G),$$

*where $E(G)$ is the number of external edges of G.*

**Proof.** — Let $\tau$ be a cyclic permutation of $[D]$ and $\mathfrak{J}_\tau(G)$ be the corresponding jacket of $G$. By Euler relation,

$$g_{\mathfrak{J}_\tau(G)} = \tfrac{1}{2}\big(2C(G) - F(\mathfrak{J}_\tau) + e(G) - V(G)\big),$$

where $e(G)$ denotes the number of internal edges of $G$. Faces of $\mathfrak{J}_\tau$ can be divided into two parts: the ones which are faces of $G$, the other ones which are not. The latter will be called external and their number will be denoted by $F_{\text{ext}}(\mathfrak{J}_\tau)$. What are these external faces exactly? Let us look at the upper left graph of Fig. 4, considered as a map. Recall that in the definition of a jacket, we removed external edges. If one does so for this map, it will contain an external face which goes all around it. This face is not a face of $G$ because it is bordered by edges of three different colours, 0, $i$ and $j$ with $i, j \neq 0$. It will be convenient to define external faces of $G$. Non-cyclic 2-bubbles $b$ of $G$ are bordered by two external vertices, namely its two vertices incident with external edges. If the edges of $b$ bear colours 0 and $i$, we call $b$ an external path of $G$ of colour $i$. An external face of $G$ of colour $ij$ is then defined as a cyclic and alternating sequence of adjacent external paths of colour $i$ and $j$ respectively. The important point to notice is that the external faces of $G$ are in bijection with the faces of $\partial G$.

External faces of $G$ of colour $ij$ are faces of $\mathfrak{J}_\tau$ if and only if $\tau$ contains the sequence $i0j$ or $j0i$. Then, a given external face of $G$ belongs to exactly $2(D-2)!$ jackets. Each face of $G$

belongs to exactly $2(D-1)!$ jackets so that

$$\frac{1}{(D-1)!} \sum_{\tau \in \mathfrak{S}_{[D]}^{(c)}} g_{\mathfrak{J}_\tau(G)} = \tfrac{1}{2(D-1)!}\big(2(D!)C(G) - 2(D-1)!F(G) - 2(D-2)!F_{\text{ext}}(G)$$

$$+ D!\big(e(G) - V(G)\big)\big)$$

$$= D\,C(G) - F(G) - \tfrac{1}{D-1}F_{\text{ext}}(G) + \tfrac{D(D-1)}{4}V(G) - \tfrac{D}{4}E(G),$$

where we used that the total number of jackets of $G$ is $D!$ and $2e(G) + E(G) = (D+1)V(G)$. Similarly, using that the total number of jackets of $\partial G$ is $(D-1)!$, that $2e(\partial G) = DV(\partial G)$ and that $V(\partial G) = E(G)$, we have

$$\frac{1}{(D-1)!} \sum_{\tau \in \mathfrak{S}_{[D]^*}^{(c)}} g_{\mathfrak{J}_\tau(\partial G)} = \tfrac{1}{2(D-1)!}\big(2(D-1)!C(\partial G) - 2(D-2)!F_{\text{ext}}(G)$$

$$+ (D-1)!\big(e(\partial G) - V(\partial G)\big)\big)$$

$$= C(\partial G) - \tfrac{1}{D-1}F_{\text{ext}}(G) + \tfrac{D-2}{4}E(G).$$

This concludes the proof. □

Coloured graphs of vanishing degree are said to be **melonic**. They form the dominant family of the $1/N$-expansion of coloured tensor models [20].

### 3.3 Intermediate field maps

The third graphical notion we will deal with corresponds to the Feynman graphs of action (2.3) *viz.,* Feynman graphs of the intermediate field representation of our model. As the intermediate field representation is a multi-matrix model, its Feynman graphs are ribbon graphs or maps. As each field $\sigma_c$ bears a colour index $c$ (and the covariance is diagonal in this colour space), the edges of these maps bear a colour too. The $\text{Tr}\log$ interaction term implies that there is no constraint on the degrees of the vertices of these maps nor on the properness of their edge-colouring. Such maps will be called **coloured maps**. As for coloured graphs, we let $\text{col}(\mathfrak{G})$ be the set of colours labelling at least one edge of $\mathfrak{G}$.

There is a bijection between the Feynman maps of the intermediate field representation and the Feynman graphs of the original tensorial action (2.2). A precise description of this bijection can be found in [63]. Let us remind the reader of its most salient features. Firstly, note that we will in fact explain a bijection between coloured graphs and coloured maps. Let $G$ be a 6-coloured graph of the $T_5^4$ model and let $\mathfrak{G}$ be the corresponding coloured map. In each $\hat{0}$-bubble of $G$, there are two sets of four parallel edges. Each set will be called a **partner link**.

Edges of $\mathfrak{G}$ are in bijection with the $\hat{0}$-bubbles (or interaction vertices) of $G$. Each such bubble has a distinguished colour, namely the colour common to the two edges which do not belong to a partner link. We label the corresponding edge of $\mathfrak{G}$ with it. Partner links of $G$ are in bijection with half-edges of $\mathfrak{G}$. Let us now describe the vertices of $\mathfrak{G}$. They form cycles of half-edges. But there is a subtlety due to external edges of $G$. Each maximal alternating sequence of adjacent 0-edges and partner links in $G$ form either a cycle or a (non cyclic) path in case of external (0-)edges. In any case, we represent such a sequence as a vertex in $\mathfrak{G}$. If a sequence is not cyclic, we add a cilium, *i.e.* a mark, to the corresponding vertex of $\mathfrak{G}$. See Fig. 5 for an illustration of this bijection.

In the sequel, we will use (at least) two features of this bijection between coloured graphs of our model and coloured maps:

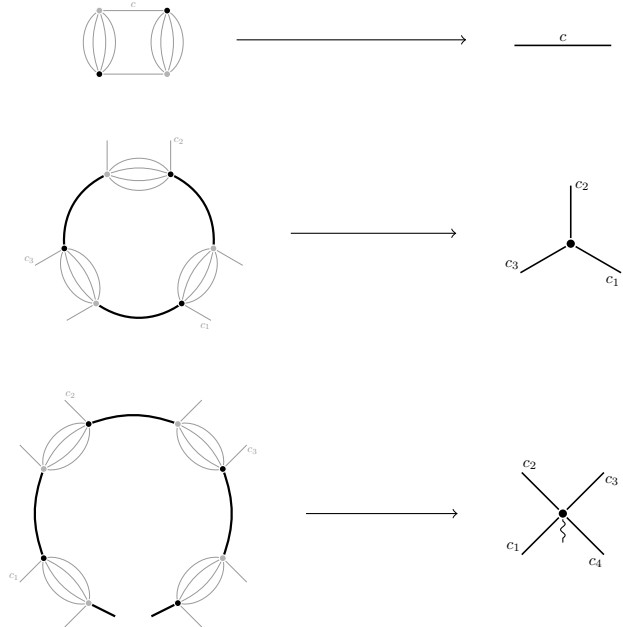

Figure 5: Bijection between coloured graphs and coloured maps – Feynman rules.

1. 0-edges correspond to corners of the coloured map [63],

2. melonic coloured graphs are in bijection with coloured plane trees [63,64].

## 4 The divergent melonic sector

### 4.1 Divergent graphs

In order to exploit the *divergence* degree

$$\omega(\mathbb{G}) = -2L(\mathbb{G}) + F(\mathbb{G})$$

of a graph $\mathbb{G}$, where $L(\mathbb{G})$ and $F(\mathbb{G})$ are respectively the number of edges and faces of $\mathbb{G}$, we need to compute the number of its faces. This quantity is given by

$$F(\mathbb{G}) = 4(V(\mathbb{G}) + 1) - 2E(\mathbb{G}) - \big(C(\partial\mathbb{G}) - 1\big) - \delta(G), \tag{4.1}$$

where $V(\mathbb{G})$ and $E(\mathbb{G})$ are respectively the number of vertices and the number of external legs of $\mathbb{G}$. Indeed, as $F(\mathbb{G}) = F_0(G) = F(G) - F_\emptyset(G)$, we can use Lemma 3.1 and the specific form of the quartic melonic interaction vertices (to compute $F_\emptyset(G) = (D-1)^2 V(\mathbb{G})$). To get eq. (4.1) we also used $V(G) = 4V(\mathbb{G})$ and set $D = 5$. The original proof of eq. (4.1) can be found in [65].

After substituting the combinatorial relation $2L + E = 4V$, the divergence degree of $\mathbb{G}$ can be written as

$$\omega(\mathbb{G}) = 4 - E(\mathbb{G}) - \big(C(\partial\mathbb{G}) - 1\big) - \delta(G). \tag{4.2}$$

**Lemma 4.1 (Superficially divergent graphs)** *The superficially divergent graphs,* i.e. *the graphs $\mathbb{G}$ such that $\omega(\mathbb{G}) \geqslant 0$, all belong to one of the cases listed in Table 1. Moreover, in the intermediate field representation,*

- *divergent four-point graphs are trees such that the unique path between their two cilia is monochrome,*

- *the closed superficially divergent graphs are*

  - *plane trees if $\omega = 5$,*
  - *unicyclic maps if $\omega = 0$ or $\omega = 2$*

*Finally, in the latter case, $\omega(\mathbb{G}) = 2$ if and only if the unique cycle of $\mathfrak{G}$ is monochrome.*

Table 1: Characteristics of superficially divergent graphs.

| $E(\mathbb{G})$ | $C(\partial\mathbb{G})$ | $\delta(G)$ | $\omega(\mathbb{G})$ |
|---|---|---|---|
| 4 | | | 0 |
| 2 | 1 | 0 | 2 |
| | | 0 | 5 |
| 0 | 0 | 3 | 2 |
| | | 5 | 0 |

**Proof.** — [59] defined two very convenient coloured graphs we will need. The first one is a chain. Chains can be broken or unbroken. In our case, chains of an intermediate field map $\mathfrak{G}$ are paths of the form $(e_1, v_1, e_2, v_2, \ldots, e_{n-1}, v_{n-1}, e_n)$ where the $e_i$'s are edges of $\mathfrak{G}$, the $v_i$'s are vertices of $\mathfrak{G}$ such that for all $i$ between 1 and $n-1$, the degree of $v_i$ in $\mathfrak{G}$ is two. Such a chain is unbroken if all its edges bear the same colour. It is broken otherwise. The second simple but very useful object is that of trivial coloured graphs or ring graphs. They consist in a single loop and no vertex. This loop bears a colour. In our case, this will always be the colour 0. Ring graphs are melonic by convention and are represented by an isolated vertex in the intermediate field representation.

According to eq. (4.2), the divergence degree of a 4-point graph $\mathbb{G}$ is bounded above by zero. It vanishes if and only if $C(\partial\mathbb{G}) = 1$ and $\delta(G) = 0$. Divergent four-point graphs are thus trees with two cilia in the intermediate field representation. Now, recursively remove all degree one vertices of this tree which do not bear a cilium. One gets a non trivial path $\mathfrak{P}$ with a cilium at each end. This path has the same power counting as the initial tree $\mathfrak{G}$. It is melonic and its boundary graph is connected if $\mathfrak{P}$ is monochrome, disconnected otherwise. Thus, according to eq. (4.2), $\mathfrak{G}$ is superficially divergent if and only if the unique path between its two cilia is monochrome.

Let us now consider a Feynman graph $\mathbb{G}$ such that $E(\mathbb{G}) = 2$. The divergence degree of such a graph is bounded above by two. The coloured extension of its boundary graph is the unique 6-coloured graph with two vertices. It is thus connected *i.e.* $C(\partial\mathbb{G}) = 1$. Then $\omega(\mathbb{G}) = 2$ if and only if $\delta(G) = 0$. Moreover, as proven in [66], if $\delta(G) > 0$ then $\delta(G) \geqslant D - 2$ where $D + 1$ is the number of colours of $G$. In our case, $D$ equals five and the smallest possible positive degree is three. Consequently the only superficially divergent 2-point graphs have vanishing degree.

Let us finally treat the case of a closed ($E = 0$) superficially divergent Feynman graph and work in the intermediate field representation. Note that the divergence degree of such a graph is bounded above by five. As a consequence, it has excess at most one. Indeed, adding an edge to a connected graph $\mathfrak{G}$ increases the number of its corners by two (hence the number of edges of $\mathbb{G}$ increases by two) while the number of faces of $\mathbb{G}$ can at most increase by one. Thus the divergence degree decreases by at least three. A connected closed graph $\mathbb{G}$ with maximal divergence degree (five) is melonic and corresponds, in the intermediate field representation,

to a tree. According to the argument above, a superficially divergent closed graph has an excess smaller or equal to one.

Let us focus on divergent graphs $\mathfrak{G}$ of excess one. They are maps with exactly two faces *i.e.* maps with a unique cycle and trees attached to the vertices of this cycle. In order to further classify such divergent graphs, as in [59], we first remove recursively all vertices of degree one. This does not change the degree of the graph. The result is a cycle $\mathfrak{C}$ *i.e.* a ring graph into which a maximal proper chain $\mathfrak{Ch}$ has been inserted. According to [59, p. 288] the Gurau degree of the coloured graph $C$ is 3 if $\mathfrak{C}$ is monochrome (the chain $\mathfrak{Ch}$ is then non-separating and unbroken with a single resulting face). It is 5 otherwise ($\mathfrak{Ch}$ is then a non-separating broken chain). $\qquad\square$

The $T_5^4$ model (2.2) has the power counting of a just renormalizable theory (and can be proven indeed perturbatively renormalizable by standard methods). However the structure of divergent subgraphs is simpler both than in ordinary $\phi_4^4$ or in the Grosse-Wulkenhaar model [72] and its translation-invariant renormalizable version.

Melonic graphs with zero, two and four external legs are divergent, respectively as $N^5$, $N^2$ and $\log N$. In the sequel we will only consider 1PI (*i.e.* one particle-irreducible or 2-edge connected) graphs as they represent the only necessary renormalizations. Melonic graphs are trees in the intermediate field representation. The condition that they are 1PI exactly corresponds to the ciliated vertices being of degree one in the tree (cilia do not count). Melonic vacuum graphs are always 1PI.

The divergent melonic graphs of the theory are obtained respectively from the fundamental melonic graphs of Fig. 6, by recursively inserting the fundamental 2-point melon on any bold line, or, in the case of the four-point function, also replacing any interaction vertex by the fundamental 4-point melon so as to create a "melonic chain" of arbitrary length (see Fig. 7 for a chain of length two), in which all vertices must be of the same colour (otherwise the graph won't be divergent).

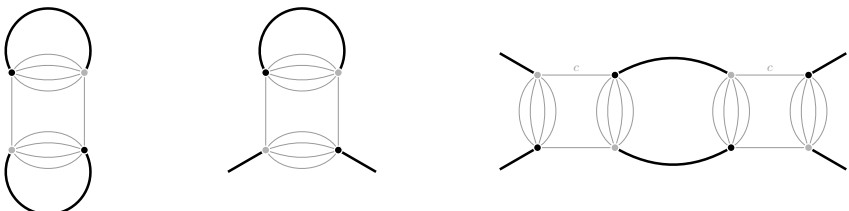

Figure 6: From left to right, the fundamental melons for the 0-, 2- and 4-point function.

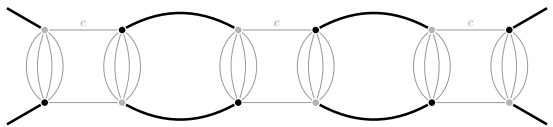

Figure 7: The length-two melonic four-point chain.

Beyond melonic approximation there is only one simple infinite family of non melonic graphs who are divergent. They are vacuum graphs diverging either as $N^2$ or as $\log N$. They are made of a "necklace chain" of arbitrary length $p \geqslant 1$, decorated with arbitrary 2-point melonic insertions. Two such necklace chains, of length one and four, are pictured in Fig. 8.

If all couplings along the chains have same colour, the divergence is quadratic, in $N^2$. If some couplings are different, the divergence is logarithmic, in $\log N$.

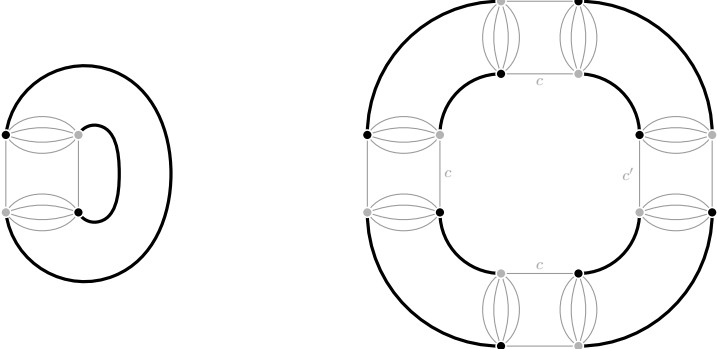

Figure 8: A length-one and a length-four non-melonic divergent vacuum connected necklaces. Remark that the left necklace diverges a $N^2$, whereas the right one diverges as $\log N$ if $c' \neq c$.

## 4.2 Melonic correlation functions

Let us call $G_{E,b}^{\mathrm{mel}}$ and $\Gamma_{E,b}^{\mathrm{mel}}$ respectively the connected and one-particle irreducible melonic functions (*i.e.* sum over the melonic Feynman amplitudes) of the theory with $E$ external fields. With a slight abuse of notation, the bare melonic two-point function $G_{2,b}^{\mathrm{mel}}(\boldsymbol{n},\overline{\boldsymbol{n}}) = \delta_{\boldsymbol{n},\overline{\boldsymbol{n}}} G_{2,b}^{\mathrm{mel}}(\boldsymbol{n})$ is related to the bare melonic self-energy $\Sigma_b^{\mathrm{mel}}(\boldsymbol{n},\overline{\boldsymbol{n}}) = \delta_{\boldsymbol{n},\overline{\boldsymbol{n}}} \Sigma_b^{\mathrm{mel}}(\boldsymbol{n})$ by the usual equation

$$G_{2,b}^{\mathrm{mel}}(\boldsymbol{n}) = \frac{C_b(\boldsymbol{n})}{1 - C_b(\boldsymbol{n})\Sigma_b^{\mathrm{mel}}(\boldsymbol{n})}.$$

$\Sigma_b^{\mathrm{mel}}(\boldsymbol{n})$ is the sum over colours $c$ of a unique (monochrome) function $\overline{\Sigma}_b^{\mathrm{mel}}$ of the single integer $n_c$:

$$\Sigma_b^{\mathrm{mel}}(\boldsymbol{n}) = \sum_c \overline{\Sigma}_b^{\mathrm{mel}}(n_c).$$

$\Sigma_b^{\mathrm{mel}}$ is uniquely defined by the last two equations and the following one (see Fig. 9)

$$\overline{\Sigma}_b^{\mathrm{mel}}(n_c) = -g_b Z_b^2 \sum_{\boldsymbol{p} \in \mathbb{Z}^5} \delta_{p_c,n_c} G_{2,b}^{\mathrm{mel}}(\boldsymbol{p}) = -g_b Z_b^2 \sum_{\boldsymbol{p} \in \mathbb{Z}^5} \frac{\delta_{p_c,n_c}}{C_b^{-1}(\boldsymbol{p}) - \Sigma_b^{\mathrm{mel}}(\boldsymbol{p})}. \tag{4.3}$$

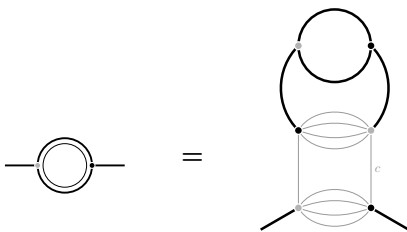

Figure 9: Pictorial representation of the relationship between $G_{2,b}^{\mathrm{mel}}$ and $\overline{\Sigma}_b^{\mathrm{mel}}$. A circle stands for a connected function, two concentric ones for a 1PI monochrome function.

Similarly the *bare* melonic four-point vertex function $\Gamma_{4,b}^{\mathrm{mel}}(\boldsymbol{n},\overline{\boldsymbol{n}},\boldsymbol{m},\overline{\boldsymbol{m}})$ is the sum over colours $c$ of contributions defined through a unique matrix $\overline{\Gamma}_{4,b}^{\mathrm{mel}}(n_c,\overline{n}_c)$ which corresponds to the melonic invariant $V_c$:

$$\Gamma_{4,b}^{\mathrm{mel}}(\boldsymbol{n},\overline{\boldsymbol{n}},\boldsymbol{m},\overline{\boldsymbol{m}}) = \sum_{c=1}^{5} \delta_{\boldsymbol{n}_{\hat{c}},\overline{\boldsymbol{n}}_{\hat{c}}} \delta_{\boldsymbol{m}_{\hat{c}},\overline{\boldsymbol{m}}_{\hat{c}}} \delta_{n_c,\overline{m}_c} \delta_{m_c,\overline{n}_c} \overline{\Gamma}_{4,b}^{\mathrm{mel}}(n_c,\overline{n}_c).$$

$\Gamma_{4,b}^{\mathrm{mel}}$ is uniquely defined by the previous equation and the following one (see Fig. 10)

$$\overline{\Gamma}_{4,b}^{\mathrm{mel}}(n_c,\overline{n}_c) = -g_b Z_b^2 \left[ 1 + \sum_{\boldsymbol{p},\overline{\boldsymbol{q}} \in \mathbb{Z}^5} \delta_{\boldsymbol{p}_{\hat{c}},\overline{\boldsymbol{q}}_{\hat{c}}} \delta_{p_c,n_c} \delta_{\overline{q}_c,\overline{n}_c} G_{2,b}^{\mathrm{mel}}(\boldsymbol{p}) G_{2,b}^{\mathrm{mel}}(\overline{\boldsymbol{q}}) \overline{\Gamma}_{4,b}^{\mathrm{mel}}(n_c,\overline{n}_c) \right],$$

which solves to

$$\overline{\Gamma}_{4,b}^{\mathrm{mel}}(n_c,\overline{n}_c) = \frac{-g_b Z_b^2}{1 + g_b Z_b^2 \sum_{\boldsymbol{p},\overline{\boldsymbol{q}}} \delta_{\boldsymbol{p}_{\hat{c}},\overline{\boldsymbol{q}}_{\hat{c}}} \delta_{p_c,n_c} \delta_{\overline{q}_c,\overline{n}_c} G_{2,b}^{\mathrm{mel}}(\boldsymbol{p}) G_{2,b}^{\mathrm{mel}}(\overline{\boldsymbol{q}})}. \tag{4.4}$$



Figure 10: Pictorial representation of the relationship between $\overline{\Gamma}_{4,b}^{\mathrm{mel}}$ and $G_{2,b}^{\mathrm{mel}}$.

At fixed cutoff $N = 2M^{j_{\max}}$, these equations define $\Sigma_b^{\mathrm{mel}}$, $G_{2,b}^{\mathrm{mel}}$ and $\Gamma_{4,b}^{\mathrm{mel}}$ (hence also $G_{4,b}^{\mathrm{mel}}$) at least as analytic functions for $g_b Z_b^2$ sufficiently small, because the species of melonic graphs is exponentially bounded as the number of vertices increases, see Section 5.4. However this does not allow to take the limit $N \to \infty$ since the radius of convergence shrinks to zero in this limit. In short we need to now renormalize.

# 5 Perturbative renormalization

## 5.1 Renormalized 1PI functions

The renormalization consists in a melonic BPHZ scheme which is given by BPHZ-like normalization conditions at zero external momenta, but restricted to the divergent sector, namely melonic graphs.[8]

The standard renormalization procedure expresses the 1PI correlation functions in terms of renormalized quantities through a Taylor expansion:

$$\Gamma_{2,b}^{\mathrm{mel}} := C_b^{-1} - \Sigma_b^{\mathrm{mel}},$$

$$\Gamma_{2,b}^{\mathrm{mel}}(\boldsymbol{n}) = \Gamma_{2,b}^{\mathrm{mel}}(0) + \Gamma_{2,mr}^{\mathrm{mel}}(\boldsymbol{n}) = \Gamma_{2,b}^{\mathrm{mel}}(0) + \boldsymbol{n}^2 \frac{\partial \Gamma_{2,b}^{\mathrm{mel}}}{\partial \boldsymbol{n}^2}(0) + \Gamma_{2,r}^{\mathrm{mel}}(\boldsymbol{n}),$$

$$\overline{\Gamma}_{4,b}^{\mathrm{mel}}(n_c,\overline{n}_c) = \overline{\Gamma}_{4,b}^{\mathrm{mel}}(0,0) + \overline{\Gamma}_{4,r}^{\mathrm{mel}}(n_c,\overline{n}_c),$$

---

[8]The true BPHZ prescription in standard field theory imposes conditions on the full 1PI functions of the theory, not just their melonic part. This is because all 1PI graphs diverge in standard field theory. In this tensorial theory since non-melonic graphs are convergent the full BPHZ prescription is not minimal, and differs from the melonic BPHZ prescription only by unnecessary finite renormalizations.

where the subscript *mr* means mass-renormalized, and *r* means full renormalization, with

$$m_r^2 = Z_b m_b^2 - \Sigma_b^{\text{mel}}(0), \quad 1 = Z_r = Z_b - \frac{\partial \overline{\Sigma}_b^{\text{mel}}}{\partial n_c^2}(0), \quad \overline{\Gamma}_{4,b}^{\text{mel}}(0,0) = -g_r Z_r^2 = -g_r.$$

Consequently we have the usual renormalization conditions

$$\Gamma_{2,mr}^{\text{mel}}(0) = \Gamma_{2,r}^{\text{mel}}(0) = 0, \quad \frac{\partial \Gamma_{2,r}^{\text{mel}}}{\partial n^2}(0) = 0, \quad \overline{\Gamma}_{4,r}^{\text{mel}}(0,0) = 0.$$

The full *melonic* two-point function is therefore

$$G_{2,b}^{\text{mel}}(\boldsymbol{n}) = \frac{\kappa_{j_{\max}}(\boldsymbol{n})}{\Gamma_{2,b}^{\text{mel}}(\boldsymbol{n})} = \frac{\kappa_{j_{\max}}(\boldsymbol{n})}{m_r^2 + \Gamma_{2,mr}^{\text{mel}}(\boldsymbol{n})} = \frac{\kappa_{j_{\max}}(\boldsymbol{n})}{\boldsymbol{n}^2 + m_r^2 + \Gamma_{2,r}^{\text{mel}}(\boldsymbol{n})},$$

so that in particular $G_{2,b}^{\text{mel}}(0) = m_r^{-2}$.

## 5.2 Mass renormalizations

Let us start by performing the mass renormalization, and postpone the wave-function and four-point coupling constant renormalization to the next section. Indeed mass renormalization is simpler as it does not involve renormalons [67]. So throughout this section we keep the bare coupling constant $g_b$, and the bare wave-function normalization $Z_b$.

The mass renormalization subtracts recursively the value of all subinsertions at 0 external momentum. Hence, recalling eq. (4.3), the monochrome melonic mass-renormalized self-energy $\overline{\Sigma}_{mr}^{\text{mel}}$ obeys the closed equation

$$\overline{\Sigma}_{mr}^{\text{mel}}(n_c) = \overline{\Sigma}_b^{\text{mel}}(n_c) - \overline{\Sigma}_b^{\text{mel}}(0) = -g_b Z_b^2 \sum_{\boldsymbol{p} \in \mathbb{Z}^5} \kappa_{j_{\max}}(\boldsymbol{p}) \frac{\delta_{p_c,n_c} - \delta_{p_c,0}}{Z_b \boldsymbol{p}^2 + m_r^2 - \sum_{c'} \overline{\Sigma}_{mr}^{\text{mel}}(p_{c'})}.$$

The sum over $\boldsymbol{p}$ in the equation above diverges only logarithmically as $j_{\max} \to \infty$. The total mass counterterm is

$$\delta_m = m_r^2 - Z_b m_b^2 = g_b Z_b^2 \sum_c \delta_m^c,$$

$$\delta_m^c = \sum_{\boldsymbol{p} \in \mathbb{Z}^5} \frac{\kappa_{j_{\max}}(\boldsymbol{p}) \delta_{p_c,0}}{Z_b \boldsymbol{p}^2 + m_r^2 - \sum_{c'} \overline{\Sigma}_{mr}^{\text{mel}}(p_{c'})} = \sum_{\boldsymbol{p} \in \mathbb{Z}^4} \frac{\kappa_{j_{\max}}(0, \boldsymbol{p})}{Z_b \boldsymbol{p}^2 + m_r^2 - \sum_{c' \neq c} \overline{\Sigma}_{mr}^{\text{mel}}(p_{c'})},$$

where we used that $\overline{\Sigma}_{mr}^{\text{mel}}(0) = 0$. Remark that $\delta_m^c$ is independent of $c$, so that in fact

$$\delta_m = 5 g_b Z_b^2 \sum_{\boldsymbol{p} \in \mathbb{Z}^4} \frac{\kappa_{j_{\max}}(0, \boldsymbol{p})}{Z_b \boldsymbol{p}^2 + m_r^2 - \sum_{c=1}^4 \overline{\Sigma}_{mr}^{\text{mel}}(p_c)}.$$

## 5.3 Effective renormalization

We want to perform only the effective (or "useful") part of the coupling constant and wave-function renormalizations, that is when the inner loop slice is higher than the external one.

### 5.3.1 Multiscale decomposition

After full mass renormalization, all correlation functions, as formal power series in $g_b$, are expressed as sums over Feynman graphs with mass-renormalized amplitudes containing mass-renormalized propagators, *i.e.*

$$C_{mr}(\boldsymbol{n}) := \frac{\kappa_{j_{\max}}(\boldsymbol{n}^2)}{Z_b \boldsymbol{n}^2 + m_r^2}.$$

Multiscale analysis amounts to decompose the (mass-renormalized here) propagator into a sum of sclice propagators $C_{mr}^{[j]}(\boldsymbol{n})$ where for each $j \in \{0, 1, \ldots, j_{\max}\}$, $C_{mr}^{[j]}(\boldsymbol{n})$ ensures that $\boldsymbol{n}^2$ is of order $M^{2j}$. Similarly to Section 2.2 we define

$$\kappa_j(\boldsymbol{n}^2) := \mathbf{1}_{[-\frac{5}{2}, \frac{5}{2}]} \star \chi_{\frac{3}{2}}(M^{-2j}\boldsymbol{n}^2) = \kappa(M^{-2j}\boldsymbol{n}^2), \quad 0 \leqslant j \leqslant j_{\max},$$

and decompose $\kappa_{j_{\max}}$ as follows

$$\kappa_{j_{\max}} = \sum_{j=0}^{j_{\max}} \eta_j, \quad \eta_j = \kappa_j - \kappa_{j-1} \text{ for } 1 \leqslant j \leqslant j_{\max}, \quad \eta_0 = \kappa_0.$$

Provided $M^2 > 2$, $\eta_j$ $(j > 0)$ is positive, smooth, and satisfies

$$\eta_j(\boldsymbol{n}^2) = \begin{cases} 0 & \text{if } \boldsymbol{n}^2 < M^{-2}M^{2j} \text{ or } \boldsymbol{n}^2 > 2M^{2j}, \\ 1 & \text{if } \frac{2}{M^2}M^{2j} \leqslant \boldsymbol{n}^2 \leqslant M^{2j}. \end{cases}$$

As a consequence, we define $C_{mr}^{[j]}(\boldsymbol{n})$ as $\eta_j(\boldsymbol{n}^2)\left(Z_b\boldsymbol{n}^2 + m_r^2\right)^{-1}$ so that $C_{mr}(\boldsymbol{n}) = \sum_{j=0}^{j_{\max}} C_{mr}^{[j]}(\boldsymbol{n})$. The decomposition of each propagator in the amplitude $A_{mr}(G)$ of any Feynman graph $G$ allows to write

$$A_{mr}(G) = \sum_{\mu \in [j_{\max}]^{e(G)}} A_{mr}^{\mu}(G).$$

$e(G)$ is the number of internal edges of $G$. $\mu$ is called a scale attribution and corresponds to choosing one index $j_\ell$ in $[j_{\max}]$ for each internal edge $\ell$ of $G$ so that the corresponding propagator is $C_{mr}^{[j_\ell]}$.

### 5.3.2 Effective constants

It will be convenient to define some more cutoff functions: for $j \in [j_{\max}]$

$$\eta_{\geqslant j} := \sum_{l=j}^{j_{\max}} \eta_l = \kappa_{j_{\max}} - \kappa_{j-1}.$$

**Definition 5.1 (Effective wave-function)** *The effective wave-function constant $Z_j$ is*

$$Z_j := Z_b - \frac{\partial \overline{\Sigma}_{mr;\geqslant j+1}^{\mathrm{mel}}}{\partial n_c^2}(0),$$

*where $\Sigma_{mr;\geqslant j}^{\mathrm{mel}}(\boldsymbol{n}) = \sum_c \overline{\Sigma}_{mr;\geqslant j}^{\mathrm{mel}}(n_c)$ is the sum of mass-renormalized amplitudes of all 1PI melonic 2-point graphs, all internal scales of which are greater than or equal to $j$, namely*

$$\overline{\Sigma}_{mr;\geqslant j}^{\mathrm{mel}}(n_c) := -g_b Z_b^2 \sum_{\boldsymbol{p} \in \mathbb{Z}^5} \eta_{\geqslant j}(\boldsymbol{p}^2) \frac{\delta_{p_c,n_c} - \delta_{p_c,0}}{Z_b \boldsymbol{p}^2 + m_r^2 - \sum_{c'} \overline{\Sigma}_{mr}^{\mathrm{mel}}(p_{c'})}.$$

Note that with these notations, $Z_{j_{\max}} = Z_b$ and $Z_{-1} = Z_r = 1$.

**Definition 5.2 (Effective coupling constant)** *The effective coupling constant $g_j Z_j^2$ is*

$$-g_j Z_j^2 := \overline{\Gamma}^{\text{mel}}_{4,b;\geqslant j+1}(0,0)$$

*where*

$$\overline{\Gamma}^{\text{mel}}_{4,b;\geqslant j}(n_c,\overline{n}_c) := \frac{-g_b Z_b^2}{1 + g_b Z_b^2 \sum_{\boldsymbol{p},\overline{\boldsymbol{q}}} \delta_{\boldsymbol{p}_{\hat{c}},\overline{\boldsymbol{q}}_{\hat{c}}} \delta_{p_c,n_c} \delta_{\overline{q}_c,\overline{n}_c} G^{\text{mel}}_{2,mr;\geqslant j}(\boldsymbol{p}) G^{\text{mel}}_{2,mr;\geqslant j}(\overline{\boldsymbol{q}})}$$

*and*

$$G^{\text{mel}}_{2,mr;\geqslant j}(\boldsymbol{n}) := \frac{\eta_{\geqslant j}(\boldsymbol{n}^2)}{Z_b \boldsymbol{n}^2 + m_r^2 - \Sigma^{\text{mel}}_{mr;\geqslant j}(\boldsymbol{n})}.$$

*With these conventions, $g_{j_{\max}} = g_b$ and $g_{-1} = g_r$.*

## 5.4 Analyticity

This section is devoted to proving that the effective wave-functions and coupling constants are analytic functions of the bare coupling $g_b$ (in a disk of radius going to 0 as $j_{\max} \to \infty$).

According to Fig. 9, the generating function for the number of 1PI divergent 2-point graphs is

$$\overline{\Sigma}^{\text{mel}}(x) = \sum_{n=1}^{\infty} 5^{n-1} C_{n-1} x^n, \qquad C_n = \frac{1}{n+1}\binom{2n}{n}.$$

This can be proven either by solving the associated equation for $\overline{\Sigma}^{\text{mel}}$,

$$\overline{\Sigma}^{\text{mel}} = \frac{x}{1 - 5\overline{\Sigma}^{\text{mel}}} \iff 5\left(\overline{\Sigma}^{\text{mel}}\right)^2 - \overline{\Sigma}^{\text{mel}} + x = 0, \tag{5.1}$$

or by noticing that divergent melonic 2-point graphs of order $n$ (the root-vertex of which has a fixed colour) are in bijection with rooted plane trees with $n-1$ edges with a choice of one colour among five per edge. As such trees are counted by the Catalan number $C_{n-1}$, we get the result.

According to eq. (4.4), the monochrome 1PI generating function $\overline{\Gamma}^{\text{mel}}_4$ of divergent 4-point graphs is given by

$$\overline{\Gamma}^{\text{mel}}_4 = x \frac{\left(1 - 5\overline{\Sigma}^{\text{mel}}\right)^2}{\left(1 - 5\overline{\Sigma}^{\text{mel}}\right)^2 - x},$$

which, from eq. (5.1), gives

$$\overline{\Gamma}^{\text{mel}}_4 = x \frac{1 - 5x - 5\overline{\Sigma}^{\text{mel}}}{1 - 6x - 5\overline{\Sigma}^{\text{mel}}}.$$

By a very simple application of the transfer theorems of Flajolet and Sedgewick [68, chapter VI], the coefficients of $\overline{\Gamma}^{\text{mel}}_4$ are asymptotically equal to $\frac{20^n}{64\sqrt{\pi n^3}}$.

Remembering the definitions of $Z_j$ and $g_j Z_j^2$ (see Section 5.3.2), we have

$$Z_j := Z_b - \frac{\partial}{\partial n_c^2} \overline{\Sigma}_{mr;\geqslant j+1}^{\text{mel}}(0) = Z_b + \sum_{n=1}^{\infty} (g_b Z_b^2)^n A_n(m_r^2, Z_b, j_{\max}, j), \qquad (5.2)$$

$$g_j Z_j^2 := -\overline{\Gamma}_{4,b;\geqslant j+1}^{\text{mel}}(0,0) = \sum_{n=1}^{\infty} (g_b Z_b^2)^n B_n(m_r^2, Z_b, j_{\max}, j). \qquad (5.3)$$

$A_n$ is the sum of the derivatives of the mass-renormalized amplitudes of the 1PI divergent melonic 2-point graphs of order $n$. $B_n$ is the sum of the mass-renormalized amplitudes of the 1PI divergent melonic 4-point graphs of order $n$. According to their generating functions, the number of such graphs is bounded by a constant to the power $n$. Moreover there certainly exist $p, q \in \mathbb{N}$ such that the amplitudes of these graphs are bounded by $(j_{\max})^{pn} M^{2qn j_{\max}}$.

Recall that

$$Z_b = 1 + \frac{\partial \overline{\Sigma}_b^{\text{mel}}}{\partial n_c^2}(0) = 1 + \frac{\partial \overline{\Sigma}_{mr}^{\text{mel}}}{\partial n_c^2}(0).$$

Then, by the implicit function theorem, $Z_b$ is an analytic function of $g_b$ in a neighbourhood of 0 (which shrinks to $\{0\}$ as $j_{\max} \to \infty$). Let us now define $F$ and $G$ on $\Omega_1 \times \Omega_2$ where $\Omega_1$ (resp. $\Omega_2$) is a complex neigbourhood of 0 (resp. of 1) such that

$$Z_j = F(g_b, Z_b) \quad \text{and} \quad g_j Z_j^2 = G(g_b, Z_b).$$

The amplitude of any divergent graph is a finite sum (because our UV cutoff is compactly supported) of analytic functions of $Z_b$ in $\Omega_2$. $A_n$ and $B_n$ are thus analytic functions of $Z_b$. Series in eqs. (5.2) and (5.3) converge normally so that $F$ and $G$ are analytic on $\Omega_1 \times \Omega_2$. Finally $Z_j$ and $g_j Z_j^2$ are holomorphic functions of $g_b$ around 0, by composition of $F$ and $G$ respectively with $Z_b(g_b)$. This proves that, at fixed UV cut-off $j_{\max}$, both $g_j Z_j^2$ and $Z_j$ are analytic functions of $g_b$ in a neighbourhood of 0.

Note also that $g_j$ is an invertible function of $g_b$ in a neighbourhood of 0.

## 5.5 Asymptotic freedom

Our aim is to prove

**Theorem 5.1** *For all $j \in \{-1, 0, \ldots, j_{max} - 1\}$,*

$$g_{j+1} - g_j = \beta_j g_j^2 + O(g_j^3),$$

*where $\beta_j = \beta_2 + O(M^{-j})$, $\beta_2$ is a negative real number and $O(g_j^3) = g_j^3 f(g_j)$ where $f$ is analytic around the origin (in a domain which shrinks to $\{0\}$ as $j_{max} \to \infty$).*

**Proof.** — Let us define $\alpha_1^{(j)}, \alpha_2^{(j)}$ and $\gamma_1^{(j)}$ as coefficients of the Taylor expansions of $g_j Z_j^2$ and $Z_j$:

$$g_j Z_j^2 =: \alpha_1^{(j)} g_b + \alpha_2^{(j)} g_b^2 + O(g_b^3), \qquad Z_j =: 1 + \gamma_1^{(j)} g_b + O(g_b^2).$$

We thus have

$$g_j = \alpha_1^{(j)} g_b + (\alpha_2^{(j)} - 2\alpha_1^{(j)} \gamma_1^{(j)}) g_b^2 + O(g_b^3) \iff g_b = \frac{1}{\alpha_1^{(j)}} g_j - \frac{1}{\left(\alpha_1^{(j)}\right)^3}(\alpha_2^{(j)} - 2\alpha_1^{(j)} \gamma_1^{(j)}) g_j^2 + O(g_j^3).$$

Inserting the previous equation into the Taylor expansion of $g_{j+1}$ at order 2, we get

$$g_{j+1} = \frac{\alpha_1^{(j+1)}}{\alpha_1^{(j)}} g_j - \frac{1}{\left(\alpha_1^{(j)}\right)^2} \left[ \frac{\alpha_1^{(j+1)}}{\alpha_1^{(j)}}(\alpha_2^{(j)} - 2\alpha_1^{(j)} \gamma_1^{(j)}) - \alpha_2^{(j+1)} + 2\alpha_1^{(j+1)} \gamma_1^{(j+1)} \right] g_j^2 + O(g_j^3).$$

Let us now compute the coefficients $\alpha_1^{(j)}, \alpha_2^{(j)}$ and $\gamma_1^{(j)}$:

$$-g_j Z_j^2 = \overline{\Gamma}_{4,b;\geqslant j+1}^{\text{mel}}(0,0) = -g_b Z_b^2 + (g_b Z_b^2)^2 \mathcal{A}_{4,2}^{(j)}(0,0) + O(g_b^3),$$

$$\mathcal{A}_{4,2}^{(j)}(n_c, \overline{n}_c) = \sum_{p \in \mathbb{Z}^4} C_{\geqslant j+1}(p, n_c) C_{\geqslant j+1}(p, \overline{n}_c),$$

where $C_{\geqslant j+1}(p) := \eta_{\geqslant j+1}(p^2)/(Z_b p^2 + m_r^2)$ and $\mathcal{A}_{4,2}^{(j)}(n_c, \overline{n}_c)$ is the mass-renormalized amplitude, "down to scale $j$", of the rightmost graph of Fig. 6. To get $\alpha_1^{(j)}$ and $\alpha_2^{(j)}$, we need the Taylor expansion of $Z_b$ at first order:

$$Z_b =: 1 + \gamma_1^{(-1)} g_b + O(g_b^2) \implies -g_j Z_j^2 = -g_b + (A_{4,2}^{(j)}(0,0) - 2\gamma_1^{(-1)})g_b^2 + O(g_b^3),$$

where $A_{4,2}^{(j)}$ equals $\mathcal{A}_{4,2}^{(j)}$ evaluated at $Z_b = 1$. We have thus

$$\alpha_1^{(j)} = 1, \quad \alpha_2^{(j)} = 2\gamma_1^{(-1)} - A_{4,2}^{(j)}(0,0).$$

Before computing the flow equation for $g_j$, we need the first order Taylor coefficient $\gamma_1^{(j)}$ of $Z_j$:

$$Z_j = Z_b - \frac{\partial \overline{\Sigma}_{mr;\geqslant j+1}^{\text{mel}}}{\partial n_c^2}(0) = Z_b + g_b Z_b^2 \frac{\partial}{\partial n_c^2} \sum_{p \in \mathbb{Z}^4} \frac{\eta_{\geqslant j+1}(p^2 + n_c^2)}{Z_b(p^2 + n_c^2) + m_r^2}\Big|_{n_c^2 = 0} + O(g_b^2)$$

$$= Z_b - g_b Z_b^3 \sum_{p \in \mathbb{Z}^4} \frac{\eta_{\geqslant j+1}(p^2)}{(Z_b p^2 + m_r^2)^2} + g_b Z_b^2 \sum_{p \in \mathbb{Z}^4} \frac{\eta_{\geqslant j+1}'(p^2)}{Z_b p^2 + m_r^2} + O(g_b^2)$$

$$=: Z_b - g_b Z_b^3 \tilde{\mathcal{A}}_{4,2}^{(j)}(0,0) + g_b Z_b^2 \mathcal{K}_j + O(g_b^2)$$

$$= 1 + \left(\gamma_1^{(-1)} - \tilde{A}_{4,2}^{(j)}(0,0) + K_j\right) g_b + O(g_b^2)$$

$$\implies \gamma_1^{(j)} = \gamma_1^{(-1)} - \tilde{A}_{4,2}^{(j)}(0,0) + K_j,$$

where, once again, $\tilde{A}_{4,2}^{(j)}$ (resp. $K_j$) equals $\tilde{\mathcal{A}}_{4,2}^{(j)}$ (resp. $\mathcal{K}_j$) evaluated at $Z_b = 1$. Finally, we get

$$g_{j+1} - g_j = -\Big[ -\left(A_{4,2}^{(j)}(0,0) - A_{4,2}^{(j+1)}(0,0)\right) + 2\left(\tilde{A}_{4,2}^{(j)}(0,0) - \tilde{A}_{4,2}^{(j+1)}(0,0)\right)$$
$$- 2K_j + 2K_{j+1}\Big]g_j^2 + O(g_j^3).$$

We now prove that $K_j$ (like $K_{j+1}$) is of order $M^{-2j}$ and that the sum of the other terms in $\beta_j$ equals a positive constant plus $O(M^{-j})$. First, we note that $\eta_j(p^2) = h(M^{-2j}p^2)$ where $h(p^2) = \kappa(p^2) - \kappa(M^2 p^2)$. Remark also that the support of $h$ is $[M^{-2}, 2]$.

$$K_j = \sum_{p \in \mathbb{Z}^4} \frac{\eta_{j+1}'(p^2)}{p^2 + m_r^2} = M^{-2(j+1)} \sum_{p \in \mathbb{Z}^4} \frac{h'(M^{-2(j+1)}p^2)}{p^2 + m_r^2}$$

$$= O(M^{-2j}) + \sum_{p \in (M^{-j-1}\mathbb{Z})^4} M^{-4(j+1)} \frac{h'(p^2)}{p^2}.$$

The above sum is a Riemann sum of the compactly supported $C^1$ function $h(p^2)/p^2$. Its difference with the corresponding integral (which vanishes) is of order of the mesh, that is $M^{-j}$.

Thus $K_j = O(M^{-j})$.

$$\tilde{A} := \tilde{A}_{4,2}^{(j)}(0,0) - \tilde{A}_{4,2}^{(j+1)}(0,0) = \sum_{p \in \mathbb{Z}^4} \frac{\eta_{j+1}(p^2)}{(p^2 + m_r^2)^2} = O(M^{-2j}) + M^{-4(j+1)} \sum_{p \in (M^{-j-1}\mathbb{Z})^4} \frac{h(p^2)}{p^4}$$

$$= O(M^{-j}) + \int_{\mathbb{R}^4} \frac{h(p^2)}{p^4} d^4p,$$

$$A := A_{4,2}^{(j)}(0,0) - A_{4,2}^{(j+1)}(0,0) = \sum_{p \in \mathbb{Z}^4} \frac{\eta_{\geqslant j+1}^2(p^2) - \eta_{\geqslant j+2}^2(p^2)}{(p^2 + m_r^2)^2}$$

$$= \sum_{p \in \mathbb{Z}^4} \frac{\eta_{j+1}^2(p^2) + 2\eta_{j+1}(p^2)\eta_{j+2}(p^2)}{(p^2 + m_r^2)^2}$$

$$= O(M^{-j}) + \int_{\mathbb{R}^4} \frac{h^2(p^2) + 2h(p^2)h(M^{-2}p^2)}{p^4} d^4p,$$

where we used $\eta_{\geqslant j+1} = \eta_{j+1} + \eta_{\geqslant j+2}$ and $\eta_i \eta_j = 0$ if $|i - j| > 1$. We get

$$\beta_j = \beta_2 + O(M^{-j}), \quad \beta_2 := -\int_{\mathbb{R}^4} \frac{d^4p}{p^4} \left[ 2h(p^2) - h^2(p^2) - 2h(p^2)h(M^{-2}p^2) \right]$$

$$= -\int_{\mathbb{R}^4} \frac{d^4p}{p^4} h(p^2) \left[ 2\left(1 - \kappa(M^{-2}p^2)\right) + \kappa(p^2) + \kappa(M^2 p^2) \right] < 0.$$

The analyticity of $g_{j+1} - g_j - \beta_j g_j^2$ as a function of $g_j$ follows from the analyticity of $g_j$ and $Z_j$ as functions of $g_b$ (see Section 5.4).  □

We have proven that for all $j$, $g_j$ is a holomorphic function of $g_b$ in a neighbourhood of 0 which goes to $\{0\}$ as $j_{\max} \to \infty$. This defines $g_{j+1}$ as a holomorphic function of $g_j$, in a neighbourhood of 0 which goes to $\{0\}$ as $j_{\max} \to \infty$. Moreover the first two coefficients of the expansion $g_{j+1}$ in powers of $g_j$ have a finite limite as $j_{\max} \to \infty$.

## 6 Holomorphic RG flow

In Section 5.5 we proved that

$$g_{j+1} = g_j + \beta_j g_j^2 + g_j^3 f(g_j) =: h_{j_{\max}, j}(g_j),$$

where $f$ is holomorphic on a neighbourhood $\Omega_{j_{\max}}$ of the origin and $\beta_j = \beta_2 + O(M^{-j})$, $\beta_2 < 0$. Note that *a priori* $\Omega_{j_{\max}} \to \{0\}$ as $j_{\max} \to \infty$. But the first two Taylor coefficients of $h_{j_{\max}, j}$ have in fact finite limits as the ultraviolet cutoff is removed. In order to know if such a result holds true at all orders, which would prove that $h_{j_{\max}, j}$ is holomorphic in a domain uniform in $j_{\max}$, we need a better understanding of the series $g_{j+1}(g_j)$. In the sequel, we assume it.

**Assumptions 1** *The series $g_{j+1}(g_j)$ is holomorphic in a domain uniform in $j_{max}$.*

The dynamics defined by $h_{j_{\max}, j}$ is not autonomous, its Taylor coefficients depend on $j$. Nevertheless, far from the infrared cutoff (here $m_r^2$), the behaviour of $\beta_j$ suggests that the dynamics becomes autonomous. In the sequel, we assume it.

**Assumptions 2** *The discrete RG flow $g_{j+1} = h(g_j)$ is defined by the iteration of a (unique) holomorphic map $h$, tangent to the identity, and such that*

$$h(z) = z + \beta_2 z^2 + O(z^3), \quad \beta_2 < 0. \tag{6.1}$$

Throughout this section, we will be interested in Cauchy problems with complex initial data. In particular, we will prove appropriate uniform boundedness of their solutions with respect to their initial data. In other words, we would like to approach results such as "for all $\epsilon > 0$, there exists a complex neighbourhood $\Omega_\epsilon$ of 0 such that $g_r = g_{-1} \in \Omega_\epsilon$ implies for all $j \geqslant 0$, $|g_j| < \epsilon$".

## 6.1 Parabolic holomorphic local dynamics

Our first objective is to understand the qualitative behaviour of the approximate RG flow (6.1) by invoking the theory of holomorphic dynamical systems. To this aim, we need to recall some classical definitions and theorems, see [69] for example.

**Definition 6.1 (Holomorphic dynamical system)** *Let $M$ be a complex manifold, and $p \in M$. A (discrete) holomorphic local dynamical system at $p$ is a holomorphic map $f : U \to M$ such that $f(p) = p$, where $U \subseteq M$ is an open neighbourhood of $p$; we shall assume that $f \neq \mathrm{id}_U$. We shall denote by $\mathrm{End}(M, p)$ the set of holomorphic local dynamical systems at $p$.*

We will only consider the case $M = \mathbb{C}$ and $p = 0$.

**Definition 6.2 (Stable set)** *Let $f \in \mathrm{End}(M, p)$ be a holomorphic local dynamical system defined on an open set $U \subseteq M$. Then the stable set $K_f$ of $f$ is*

$$K_f := \bigcap_{k=0}^{\infty} f^{\circ(-k)}(U).$$

In other words, the stable set of $f$ is the set of all points $z \in U$ such that the orbit $\{f^{\circ k}(z) : k \in \mathbb{N}\}$ is well-defined. If $z \in U \setminus \{K_f\}$, we shall say that $z$ (or its orbit) escapes from $U$. Clearly, $p \in K_f$ and so the stable set is never empty (but it can happen that $K_f = \{p\}$).

**Definition 6.3 (Conjugation)** *We say that $f, g \in \mathrm{End}(\mathbb{C}, 0)$ are holomorphically conjugated if there exists a holomorphic map $h$ such that $h \circ f = g \circ h$.*

**Definition 6.4 (Classification)** *Let $f \in \mathrm{End}(\mathbb{C}, 0)$ be given by $f(z) = \lambda z + \sum_{j \geqslant 2} a_j z^j$. We say that $f$ is*

1. *hyperbolic if $|\lambda| \neq 1$,*

2. *parabolic if $\lambda^q = 1$ for some $q \in \mathbb{N} \setminus \{0\}$,*

3. *elliptic if $|\lambda| = 1$ and $\lambda^q \neq 1$ for all $q \in \mathbb{N} \setminus \{0\}$.*

The RG flow we consider here is thus a parabolic dynamical system ($\lambda = 1$).

**Definition 6.5 (Multiplicity)** *Let $f \in \mathrm{End}(\mathbb{C}, 0)$ be a holomorphic local dynamical system with a parabolic fixed point at the origin. Then we can write:*

$$f(z) = e^{2i\pi p/q} z + a_{r+1} z^{r+1} + O(z^{r+2}),$$

*with $a_{r+1} \neq 0$. $r + 1$ is called the multiplicity of $f$.*

**Definition 6.6 (Directions)** *Let $f \in \mathrm{End}(\mathbb{C}, 0)$ be tangent to the identity of multiplicity $r+1 \geqslant 2$. Then a unit vector $v \in \mathbb{S}^1$ is an attracting (resp. repelling) direction for $f$ at the origin if $a_{r+1} v^r$ is real negative (resp. real positive).*

Clearly, there are $r$ equally spaced attracting directions, separated by $r$ equally spaced repelling directions: if $a_{r+1} = |a_{r+1}|e^{i\alpha}$, then $v = e^{i\theta}$ is attracting (resp. repelling) if and only if

$$\theta = \frac{2k+1}{r}\pi - \frac{\alpha}{r} \quad \left(\text{resp. } \theta = \frac{2k}{r}\pi - \frac{\alpha}{r}\right).$$

It turns out that to every attracting direction is associated a connected component of $K_f \setminus \{0\}$.

**Definition 6.7 (Basins)** *Let $v \in \mathbb{S}^1$ be an attracting direction for an $f \in \text{End}(\mathbb{C}, 0)$ tangent to the identity. The basin centerd at $v$ is the set of points $z \in K_f \setminus \{0\}$ such that $\lim_{k\to\infty} f^{\circ k}(z) = 0$ and $\lim_{k\to\infty} f^{\circ k}(z)/|f^{\circ k}(z)| = v$. If $z$ belongs to the basin centered at $v$, we shall say that the orbit of $z$ tends to $0$ tangent to $v$.*

**Definition 6.8 (Petals)** *An attracting petal centered at an attracting direction $v$ of an $f \in \text{End}(\mathbb{C}, 0)$ tangent to the identity is an open simply connected $f$-invariant set $P \subseteq K_f \setminus \{0\}$ such that a point $z \in K_f \setminus \{0\}$ belongs to the basin centered at $v$ if and only if its orbit intersects $P$. In other words, the orbit of a point tends to $0$ tangent to $v$ if and only if it is eventually contained in $P$. A repelling petal (centered at a repelling direction) is an attracting petal for the inverse of $f$.*

**Theorem 6.1 (Leau-Fatou flower)** *Let $f \in \text{End}(\mathbb{C}, 0)$ be a holomorphic local dynamical system tangent to the identity with multiplicity $r + 1 \geqslant 2$ at the fixed point. Let $v_1^\pm, \ldots, v_r^\pm \in \mathbb{S}^1$ be the attracting (resp. repelling) directions of $f$ at the origin. Then,*

1. *for each attracting (resp. repelling) direction $v_j^\pm$ there exists an attracting (resp. repelling petal) $P_j^\pm$, so that the union of these $2r$ petals together with the origin forms a neighbourhood of the origin. Furthemore, the $2r$ petals are arranged cyclically so that two petals intersects if and only if the angle between their central directions is $\pi/r$.*

2. *$K_f \setminus \{0\}$ is the (disjoint) union of the basins centered at the $r$ attracting directions.*

3. *If $B$ is a basin centered at one of the attracting directions, then there is a function $\varphi : B \to \mathbb{C}$ such that $\varphi \circ f(z) = \varphi(z) + 1$ for all $z \in B$. Furthermore if $P$ is the corresponding petal, then $\varphi|_P$ is a biholomorphism with an open subset of the complex plane containing a right-half plane – and so $f|_P$ is holomorphically conjugated to the translation $z \mapsto z + 1$.*

As a consequence of Theorem 6.1, if $z$ belongs to an attracting petal $P$ of a holomorphic local dynamical system tangent to the identity, then its entire orbit is contained in $P$ and moreover $f^{\circ n}(z)$ goes to $0$ (as $n \to \infty$), tangentially to the corresponding attracting direction. A typical trajectory can be seen on Fig. 11. Note that Theorem 6.1 asserts the existence of attracting and repelling petals whose union with the origin forms a neighbourhood of the origin. The intersection properties of these petals implies that their asymptotic opening angle (*i.e.* their opening angle close to 0) is strictly bigger than $\pi/r$ for a system of multiplicity $r$. But in fact, with a bit more work, one can construct petals whose asymptotic opening angle is $2\pi/r$, see [70]. Such attracting petals are tangent at 0 to their two neighbouring repelling directions.

In case of the system (6.1), we have a parabolic dynamical system of multiplicity 2 so that there is only one attracting (resp. repelling) petal corresponding to the attractive (resp. repelling) direction $(1,0)$ (resp. $(-1,0)$). The asymptotic opening angle of the attracting petal is $2\pi$ which makes it very similar to cardioid-like domains obtained by Loop Vertex Expansion [46,47], see Fig. 12.

In the next sections, we get more quantitative results on the RG trajectories in case $g_r$ is real, on the shapes of attracting petals, and on the size of the Nevanlinna-Sokal disk they

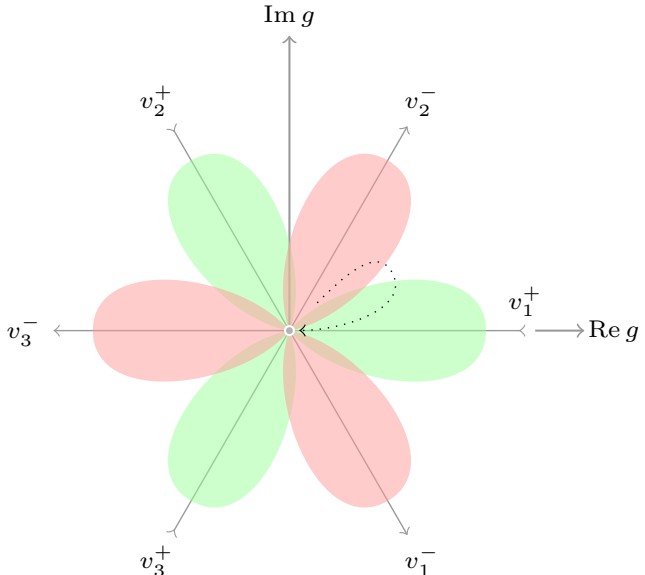

Figure 11: Attracting (green) and repelling (red) petals of a dynamics of multiplicity 4, and a typical trajectory.

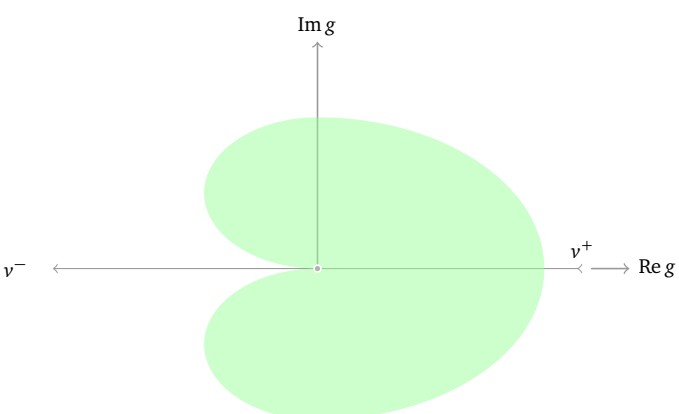

Figure 12: A unique attracting petal of a multiplicity 2 parabolic dynamics.

contain. To this aim, we will study continuous dynamical systems, more precisely first-order ODEs, rather than iterations of holomorphic maps. This is justified by the following argument. As we saw in Section 5.5, there is evidence that the discrete RG flow of the $T_5^4$ tensor field is well approximated by the dynamical system (6.1), at least in the deep ultraviolet. From Theorem 6.1 a trajectory starting in the unique attracting petal $P_+$ remains forever in this petal. Moreover, in $P_+$, the dynamics is conjugated to the translation $z \mapsto z+1$. But this translation is the time 1 flow of the constant vector field $Y = \partial_z$. Thus there exists a holomorphic vector field $X$ such that $h$ in eq. (6.1) is the time 1 flow of $X$. The Taylor coefficients of $X$ can be computed recursively via the equation $e^X(z) = h(z)$. One finds $X = \left( \beta_2 z^2 + (\beta_3 - 2\beta_2^2)z^3 + O(z^4) \right) \partial_z$ if $h(z) = z + \beta_2 z^2 + \beta_3 z^3 + O(z^4)$. As a consequence we will consider, in the next sections, ODEs of the form

$$g' = \beta_2 g^2 + \beta_3 g^3 + O(g^4), \quad \beta_2 \in \mathbb{R}_-,$$

keeping in mind that the above $\beta_3$ corresponds in fact to $\beta_3 - 2\beta_2^2$ in the notation of eq. (6.1).

## 6.2 Quadratic flow

Let us first consider $f : \mathbb{R} \times \mathbb{C} \to \mathbb{C}$ such that $f(t, z) = \beta_2 z^2$, $\beta_2$ real negative, and the following Cauchy problem:

$$g' = f(t, g), \qquad (6.2a)$$
$$g(0) = g_r \in \mathbb{C}. \qquad (6.2b)$$

The solution is obviously given by

$$\frac{1}{g(t)} = \frac{1}{g_r} - \beta_2 t.$$

In polar coordinates ($g = \rho e^{i\theta}$, $\theta \in [-\pi, \pi]$), we have

$$\frac{1}{g(t)} = \frac{e^{-i\theta}}{\rho} = \frac{\cos \theta_r}{\rho_r} - \beta_2 t - i \frac{\sin \theta_r}{\rho_r}. \qquad (6.3)$$

If $\theta_r \in (-\pi, \pi) \setminus \{0\}$, $g$ is well-defined on $\mathbb{R}$. If $\theta_r = 0$, $g$ is well-defined on $\mathbb{R}_+$ (but explodes at a finite negative time). If $\theta_r = \pi$, $g$ explodes at a finite (positive) time.

We now prove a uniform bound on $|g|$ for $g_r$ in the following compact domain $\Omega_\epsilon$ of the complex plane.

**Definition 6.9 (Domain of uniform boundedness of a quadratic flow)** *Let $\epsilon$ be a positive real number. In polar coordinates ($z = \rho e^{i\theta}$, $\rho \in \mathbb{R}_+$, $\theta \in [-\pi, \pi]$),*

$$\Omega_\epsilon := \begin{cases} \{z \in \mathbb{C} : \rho \leqslant \epsilon\} & \text{if } \theta \in [-\frac{\pi}{2}, \frac{\pi}{2}], \\ \{z \in \mathbb{C} : \rho \leqslant \epsilon |\sin \theta|\} & \text{if } |\theta| \in (\frac{\pi}{2}, \pi]. \end{cases}$$

The set $\Omega_\epsilon$ is made of three parts. On $\{\operatorname{Re} z \geqslant 0\}$, $\Omega_\epsilon$ is a closed half-disk of radius $\epsilon$, centered at 0. On $\{\operatorname{Re} z \leqslant 0\} \cap \{\operatorname{Im} z \geqslant 0\}$, $\Omega_\epsilon$ is a closed half-disk of radius $\frac{\epsilon}{2}$ centered at $i\frac{\epsilon}{2}$. On $\{\operatorname{Re} z \leqslant 0\} \cap \{\operatorname{Im} z \leqslant 0\}$, $\Omega_\epsilon$ is a closed half-disk of radius $\frac{\epsilon}{2}$ centered at $-i\frac{\epsilon}{2}$. See Fig. 13 for a picture of $\Omega_\epsilon$.

**Remark.** — $\Omega_\epsilon$ contains the cardioid domain $\mathscr{C}_\epsilon := \left\{ \rho \leqslant \epsilon \cos^2(\frac{\theta}{2}) \right\}$ which is the typical domain of analyticity of correlation functions predicted by Loop Vertex Expansion.

**Theorem 6.2** *If $g_r \in \Omega_\epsilon$, then for all $t \in \mathbb{R}_+$, $|g(t)| \leqslant \epsilon$.*

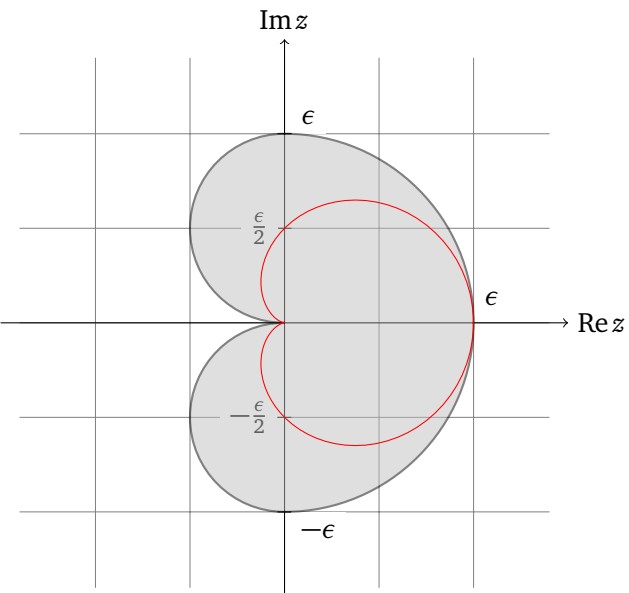

Figure 13: In gray, the domain $\Omega_\epsilon$ of Definition 6.9. In red, the cardioid $\rho = \epsilon \cos^2(\frac{\theta}{2})$.

**Proof.** — From eq. (6.3),

$$\frac{1}{\rho^2(t)} = \left( \frac{\cos \theta_r}{\rho_r} - \beta_2 t \right)^2 + \frac{\sin^2 \theta_r}{\rho_r^2}. \tag{6.4}$$

If $\theta_r \in [-\frac{\pi}{2}, \frac{\pi}{2}]$, $\cos \theta_r \geqslant 0$ and $\rho$ attains its maximum at $t = 0$ (recall that $\beta_2 < 0$) so that $\rho(t) \leqslant \rho_r$. If $|\theta_r| \in (\frac{\pi}{2}, \pi)$, $\cos \theta_r < 0$ and $\rho(t) \leqslant \frac{\rho_r}{|\sin \theta_r|}$. This proves the desired bound. $\qquad\square$

**Remark.** — In fact, by the holomorphic (on $\mathbb{C}^*$) change of coordinate $z \mapsto 1/z$, one can even prove that $g_r \in \Omega_\epsilon$ implies $g(t) \in \Omega_\epsilon$ for all $t > 0$.

## 6.3 Cubic flow

We now consider the following complex cubic differential flow:

$$g' = \beta_2 g^2 + \beta_3 g^3 = \beta_2(x^2 - y^2) + \beta_3(x^3 - 3xy^2) + 2i\beta_2 xy + i\beta_3(3x^2 y - y^3), \tag{6.5a}$$
$$g(0) = g_r \in \mathbb{C}, \tag{6.5b}$$

where $x = \mathrm{Re}(g)$ and $y = \mathrm{Im}(g)$, $\beta_2, \beta_3$ real with $\beta_2 < 0$. We have

**Theorem 6.3** *For $\epsilon$ small enough, if $g_r \in \Omega_\epsilon$ then there exists a function*

$$\phi : \mathbb{R}_+ \times \Omega_\epsilon \to \mathbb{C}$$
$$(t, g_r) \mapsto \phi(t, g_r)$$

*holomorphic in $g_r$, uniformly bounded, namely*

$$|\phi(t, g_r)| < 2\pi, \quad \text{for all } t \in \mathbb{R}_+ \text{ and } g_r \in \Omega_\epsilon,$$

*such that the unique maximal solution of the Cauchy problem eq. (6.5) defined on $\mathbb{R}_+$ is*

$$g(t) = \frac{g_r}{1 - \beta_2 g_r t + \frac{\beta_3}{\beta_2} g_r \log(1 - \beta_2 g_r t) + \frac{\beta_3}{\beta_2} g_r \phi(t)}.$$

**Proof.** — The partial derivatives of the right-hand side of eq. (6.5a) with respect to $x$ and $y$ are continuous so that the Cauchy-Lipschitz theorem applies. As a consequence, for any complex initial data $g_r$, there exists a unique maximal continuously differentiable solution $g$ defined on $[0, T)$ for some $T > 0$. Moreover if $\text{Im}\, g_r$ is positive (resp. negative), then for all $t \in [0, T)$, $\text{Im}\, g(t)$ is positive (resp. negative). In particular $g(t) \neq 0$.

Let $g_2$ and $g_3$ be the two following complex functions on $\mathbb{R}_+$:

$$g_2(t) = g_r \left(1 - \beta_2 g_r t\right)^{-1},$$
$$g_3(t) = g_r \left(1 - \beta_2 g_r t + \frac{\beta_3}{\beta_2} g_r \log(1 - \beta_2 g_r t)\right)^{-1}.$$

$g_2$ is a solution of $g' = \beta_2 g^2$ and $g_3$ a solution of $g' = \beta_2 g^2 + \beta_3 g_2 g^2$. Let us define the following new variables:

$$u := \frac{g_r}{g}, \quad u_3 := \frac{g_r}{g_3}, \quad \alpha := |\beta_2| g_r, \quad \beta := \frac{\beta_3}{\beta_2} g_r.$$

In these new variables, eq. (6.5a) rewrites as

$$u' = \alpha \left(1 + \frac{\beta}{u}\right). \tag{6.6}$$

Let us insert the ansatz $g^{-1} = g_3^{-1} + \frac{\beta_3}{\beta_2} \phi$ (or $u = u_3 + \beta \phi$) into eq. (6.6) to get the ODE satisfied by $\phi$:

$$\phi' = -\frac{\alpha}{1 + \alpha t} \frac{\beta \log(1 + \alpha t) + \beta \phi}{u_3 + \beta \phi}. \tag{6.7}$$

The function $g$ of Theorem 6.3 is a solution of the Cauchy problem (6.5) if and only if $\phi$ satisfies (6.7) with initial condition $\phi(0) = 0$.

Let $D$ be the following open subset of $\mathbb{R} \times \mathbb{C}$:

$$D := \left\{(t, z) \in (-\tfrac{1}{\alpha}, +\infty) \times \mathbb{C} : u_3(t) + \beta z \neq 0\right\}.$$

Let $h$ be the function defined by:

$$h : D \to \mathbb{C}$$
$$(t, z) \mapsto -\frac{\alpha}{1 + \alpha t} \frac{\beta \log(1 + \alpha t) + \beta z}{u_3 + \beta z}.$$

It is easy to check that the partial derivatives of $h$ are continuous on $D$ so that $h$ is continuously differentiable on $D$. Then, as $(0, 0) \in D$, by the Cauchy-Lipschitz theorem, there exists a unique (continuously differentiable) solution $\phi$ to $\phi' = h(t, \phi)$ such that $\phi(0) = 0$. In particular, $\phi$ is defined on $[0, T)$ for some $T > 0$.

Let us now show that $\phi$ is bounded (and thus defined in fact on $\mathbb{R}_+$). We integrate eq. (6.6) out: for $t \in [0, T)$,

$$\int_0^t \frac{u u'}{u + \beta} \, dt = \int_u \frac{z}{z + \beta} \, dz = u - 1 - \beta \log\left(\frac{u + \beta}{1 + \beta}\right) = \alpha t,$$

where we used $u(0) = 1$. This implicit equation for $u$ can be turned into an implicit equation for $\phi$ under the ansatz $u = u_3 + \beta\phi$:

$$\phi = \log\left(\frac{u_3 + \beta\phi + \beta}{(1+\beta)(1+\alpha t)}\right). \tag{6.8}$$

If there exists $\tau \in [0, T)$ such that $|\phi(\tau)| > 2\pi$, we define $t_0$ as the smallest $t \in [0, T)$ such that $|\phi(t)| = 2\pi$. Thus on $[0, t_0]$, the continuous function $|\phi|$ is smaller than $2\pi$ and must take all values between $0$ and $2\pi$. But,

$$\phi = -\log(1+\beta) + \log\left(1 + \beta\frac{\log(1+\alpha t) + \phi + 1}{1+\alpha t}\right),$$

$$|\phi| \leqslant |\log(1+\beta)| + \left|\log\left(\left|1 + \beta\frac{\log(1+\alpha t) + \phi + 1}{1+\alpha t}\right|\right)\right| + \pi$$

$$\leqslant |\log(1+\beta)| + \left|\log\left(1 + |\beta|\frac{|\log(1+\alpha t)| + |\phi+1|}{|1+\alpha t|}\right)\right| + \pi.$$

If $|g_r|$ is small enough, $|1+\beta|$ is close to $1$ and $\arg(1+\beta)$ is small so that $|\log(1+\beta)|$ can be made smaller than $\frac{1}{2}$ (say). Let us recall that in polar coordinates, $g_r = \rho_r e^{i\theta_r}$. A simple inspection of $|1+\alpha t|$ as a function of $t \in \mathbb{R}_+$ shows that

$$|1+\alpha t| \geqslant \begin{cases} 1 & \text{if } |\theta_r| \in [0, \frac{\pi}{2}], \\ |\sin\theta_r| & \text{if } |\theta_r| \in (\frac{\pi}{2}, \pi). \end{cases} \tag{6.9}$$

As a consequence, for $g_r \in \Omega_\epsilon$ and $\epsilon$ small enough,

$$|\beta|\frac{|\phi+1|}{|1+\alpha t|} \leqslant \left|\frac{\beta_3}{\beta_2}\right|\frac{|g_r|}{|\sin\theta_r|}(|\phi|+1) \leqslant \left|\frac{\beta_3}{\beta_2}\right|(2\pi+1)\epsilon \leqslant \frac{1}{3}.$$

There remains to bound

$$|\beta|\frac{|\log(1+\alpha t)|}{|1+\alpha t|} \leqslant |\beta|\frac{\frac{1}{2}|\log|1+\alpha t|^2| + \pi}{|1+\alpha t|}.$$

Firstly,

$$|\beta|\frac{\pi}{|1+\alpha t|} \leqslant \pi\left|\frac{\beta_3}{\beta_2}\right|\frac{|g_r|}{|\sin\theta_r|} \leqslant \frac{1}{3}.$$

Secondly, let us define the functions $f, g : [0, T) \to \mathbb{R}$ by $f(t) = |1+\alpha t|^2$ and $g(t) = \frac{\log f(t)}{\sqrt{f(t)}}$. From

$$f'(t) = 2\rho_r|\beta_2|\cos\theta_r + 2t\rho_r^2|\beta_2|^2 \quad \text{and} \quad g'(t) = \frac{f'(t)}{2f(t)^{3/2}}\left(2 - \log f(t)\right),$$

we deduce that $0 \leqslant g(t) \leqslant \frac{2}{e}$ so that

$$\frac{|\beta|}{2}\frac{|\log|1+\alpha t|^2|}{|1+\alpha t|} \leqslant \left|\frac{\beta_3}{\beta_2}\right|\frac{|g_r|}{e} \leqslant \frac{1}{3}.$$

Thirdly,

$$|\phi| \leqslant \frac{1}{2} + \pi + \log 2 < 2\pi,$$

which is a contradiction and proves that $\left\|\phi\right\|_\infty < 2\pi$. Finally, to prove that $\phi$ is a holomorphic function of $g_r$, we note that it is a solution of the implicit equation

$$F_t(g_r, z) = \frac{u_3 + \beta z + \beta}{1 + \beta} - (1 + \alpha t)e^z = 0$$

(see eq. (6.8)). But $F_t$ is holomorphic on $\left(\mathbb{C} \setminus \{-\frac{\beta_2}{\beta_3}\}\right) \times \mathbb{C}$ so that for $g_r \in \Omega_\epsilon$ and $\epsilon$ small enough, $\phi$ is holomorphic in $g_r$ by the implicit function theorem. $\square$

Let $\epsilon$ be a positive real number. Denoting the ratio $\beta_3/\beta_2$ by $\beta_{3,2}$, we define $\mathcal{H}_\epsilon$ as the following compact subset of $\mathbb{C}$:

$$\mathcal{H}_\epsilon := \begin{cases} \left\{z \in \mathbb{C} : |z| \leqslant \frac{\epsilon}{1 + 3\pi|\beta_{3,2}|\epsilon}\right\} & \text{if } |\arg z| \in [0, \frac{\pi}{2}], \\ \left\{z \in \mathbb{C} : |z| \leqslant \frac{\epsilon|\sin\arg z|}{1 + \epsilon|\beta_{3,2}|(|\log|\sin\arg z|| + 3\pi)}\right\} & \text{if } |\arg z| \in (\frac{\pi}{2}, \pi). \end{cases}$$

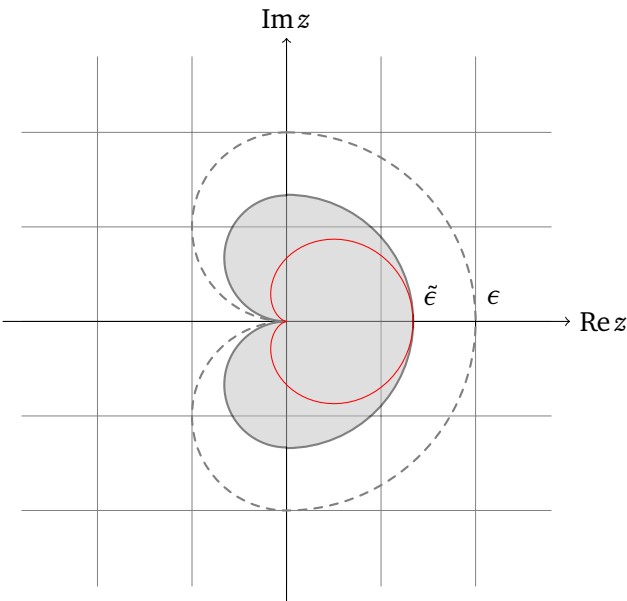

Figure 14: In gray, the domain $\mathcal{H}_\epsilon$ ($\tilde{\epsilon} := \frac{\epsilon}{1 + 3\pi|\beta_{3,2}|\epsilon}$, here $3\pi|\beta_{3,2}| = \frac{1}{2}$). The red (resp. dashed) line is the boundary of the cardioid $\mathscr{C}_{\tilde{\epsilon}}$ (resp. of $\Omega_\epsilon$).

**Corollary 6.1** *Let $\epsilon$ be smaller or equal to $1/|\beta_{3,2}|$. If $g_r \in \mathcal{H}_\epsilon$ then the solution of the Cauchy problem eq. (6.5) given in Theorem 6.3 is bounded above (in modulus) by $\epsilon$.*

**Proof.** — Let us denote by $f$ the function from $\mathbb{R}_+$ to $\mathbb{R}_+$ defined by $f(t) = |1 + \alpha t|$. By Theorem 6.3,

$$|g(t)| = \frac{|g_r|}{|1 + \alpha t + \beta_{3,2}g_r\log(1 + \alpha t) + \beta_{3,2}g_r\phi(t)|},$$

where $\beta_{3,2} = \beta_3/\beta_2$. But

$$\left|1 + \alpha t + \beta_{3,2}g_r\log(1 + \alpha t) + \beta_{3,2}g_r\phi(t)\right| \geqslant |1 + \alpha t| - |\beta_{3,2}g_r|\left(|\log(1 + \alpha t)| + |\phi(t)|\right)$$
$$\geqslant |1 + \alpha t| - |\beta_{3,2}g_r|\left(|\log|1 + \alpha t|| + 3\pi\right). \quad (6.10)$$

Let us assume for a moment that this last lower bound is non-negative. Then,

$$|g(t)| \leqslant \frac{|g_r|}{|1+\alpha t| - |\beta_{3,2} g_r| \left( \left| \log|1+\alpha t| \right| + 3\pi \right)} \leqslant \epsilon \tag{6.11}$$

if and only if

$$|g_r| \leqslant \frac{\epsilon |1+\alpha t|}{1 + \epsilon |\beta_{3,2}| \left( \left| \log|1+\alpha t| \right| + 3\pi \right)}. \tag{6.12}$$

Now, the right-hand side of eq. (6.10) is non-negative if and only if

$$|g_r| \leqslant \frac{|1+\alpha t|}{|\beta_{3,2}| \left( \left| \log|1+\alpha t| \right| + 3\pi \right)}. \tag{6.13}$$

Consequently, if we impose eq. (6.12), eq. (6.13) is satisfied and this justifies *a posteriori* the first inequality of eq. (6.11).

Finally, let $g$ be the function from $\mathbb{R}_+$ into itself defined by $g(x) = \frac{\epsilon x}{1 + \epsilon |\beta_{3,2}| (|\log x| + 3\pi)}$. It is easy to check that if $\epsilon \leqslant 1/|\beta_{3,2}|$, $g$ is increasing. As a consequence, from eq. (6.9),

$$|g_r| \leqslant \begin{cases} \frac{\epsilon}{1 + 3\pi |\beta_{3,2}| \epsilon} & \text{if } \theta_r \in [0, \frac{\pi}{2}], \\ \frac{\epsilon |\sin \theta_0|}{1 + \epsilon |\beta_{3,2}| (|\log|\sin \theta_r|| + 3\pi)} & \text{if } \theta_r \in [\frac{\pi}{2}, \pi), \end{cases}$$

implies eq. (6.12) and $|g(t)| \leqslant \epsilon$. $\qquad \square$

## 6.4 Differential flow of higher degree

Let us now consider more general complex differential equations and prove that for sufficiently small initial conditions, their solutions are uniformly bounded as well. Let $U$ be a complex neighbourhood of 0. Let $f$ be the following function:

$$\begin{aligned} f : \mathbb{R}_+ \times U &\to \mathbb{C} \\ (t, z) &\mapsto \beta_2 z^2 + \beta_3 z^3 + z^4 h(z), \end{aligned} \tag{6.14}$$

where $h$ is holomorphic on $U$. We are interested in the following Cauchy problem:

$$g' = f(t, g), \tag{6.15a}$$

$$g(0) = g_r \in \mathbb{C}. \tag{6.15b}$$

**Definition 6.10 (Disks)** *Let $r$ be real and positive. We will denote by $\mathbb{D}_r$ the open disk of radius $r$ centered at 0. An open disk $\mathbb{S}_r$ of radius $r$ centered at $r$ will be called a Nevanlinna-Sokal disk. $\mathbb{S}_r$ is the set of complex numbers $z$ such that $\mathrm{Re}\left( \frac{1}{z} \right) > \frac{1}{2r}$ or equivalently $|z| < 2r \cos(\arg z)$.*

**Theorem 6.4** *Let $\epsilon$ be a sufficiently small positive real number. There exists a simply connected domain $D_\epsilon$ of $\mathbb{C}$ such that $D_\epsilon \subset U$, $0 \in \partial D_\epsilon$, and $D_\epsilon$ contains a Nevanlinna-Sokal disk $\mathbb{S}_\delta$, $\delta = \frac{1}{6} \frac{\epsilon}{1 + \frac{3\pi}{2} |\beta_{3,2}| \epsilon}$, such that if $g_r \in D_\epsilon$ then, for all $t \geqslant 0$, the unique maximal solution of the Cauchy problem eq. (6.15) defined on $\mathbb{R}_+$ belongs to $\mathbb{D}_\epsilon$.*

**Proof.** — By the Leau-Fatou flower theorem, there exists an attracting petal directed along the positive real axis, see Fig. 12. Consequently we know that there exists an open connected and simply connected complex subset as claimed in Theorem 6.4. But we have no control on the size of the Nevanlinna-Sokal disk it contains. Thus we follow a more pedestrian road.

By a biholomorphic change of variable $y = \varphi(g)$, eq. (6.15a) rewrites as

$$y' = -y^2 + \frac{\beta_3}{\beta_2^2} y^3.$$

This comes from a theorem of G. Szekeres, the proof of which can be found in [71]. Let us briefly repeat the arguments here. Let $U$ be a complex neighbourhood of 0 and $f$ a holomorphic function on $U$. The corresponding ODE writes

$$z' = f(z) \iff \frac{dz}{f(z)} = dt.$$

The meromorphic differential form $\omega = \frac{dz}{f(z)}$ is the dual form of the vector field $f(z)\partial_z$. Let us assume that 0 is a pole of $\omega$ of order $p + 1$, $p \geqslant 1$, and note the corresponding residue $a_{-1}$. In the case of the vector field eq. (6.14), $p = 1$ and $a_{-1} = -\frac{\beta_3}{\beta_2^2}$.

Let us explain how to prove that

$$\omega = \frac{dy}{-y^{p+1} - a_{-1}y^{2p+1}}$$

after a biholomorphic change of coordinates $y = \varphi(z)$. Firstly, as $1/f$ has a pole of order $p+1$ at 0, there exists a holomorphic function $v$ (near 0) such that

$$\omega = \frac{a_{-1}\,dz}{z} + d\left(\frac{v}{z^p}\right), \quad v(0) \neq 0.$$

Secondly, by integrating the equality

$$\frac{a_{-1}\,dz}{z} + d\left(\frac{v}{z^p}\right) = \frac{dy}{-y^{p+1} - a_{-1}y^{2p+1}},$$

one obtains an implicit equation for $y$:

$$a_{-1}\log z + \frac{v(z)}{z^p} = \frac{1}{py^p} + a_{-1}\log y - \frac{a_{-1}}{p}\log(1 + a_{-1}y^p). \tag{6.16}$$

Thirdly, defining $y = \varphi(z) =: zu(z)$, eq. (6.16) becomes

$$\frac{1}{pu^p} + a_{-1}z^p\log u - \frac{a_{-1}}{p}z^p\log\left(1 + a_{-1}(zu)^p\right) - v(z) = 0,$$

and by the implicit function theorem, there exists a neighbourhood $V$ of 0 on which $u$ (then $\varphi$) is holomorphic ($\varphi$ is even biholomorphic because $\varphi'(0) = u(0) \neq 0$).

From Theorem 6.3, there exists a unique maximal solution $y(t)$ of the Cauchy problem

$$y' = -y^2 + \frac{\beta_3}{\beta_2^2} y^3, \quad y(0) = y_0,$$

which is defined on $\mathbb{R}_+$. Moreover, from Corollary 6.1, if $r'$ is smaller than $\frac{\beta_2^2}{|\beta_3|}$, then if $y_0 \in \mathcal{H}_{r'}$, $y(t) \in \mathbb{D}_{r'}$ for all $t \geqslant 0$. If $r'$ is small enough then $\overline{\mathbb{D}}_{r'}$ and thus $\overline{\mathcal{H}}_{r'}$ are subsets of $V$. As a

consequence, if $g_r \in \varphi^{-1}(\mathcal{H}_{r'}) =: \Omega$ then $g(t) = \varphi^{-1}\big(y(t)\big)$ is the unique maximal solution of eq. (6.15) defined on $\mathbb{R}_+$ and for all $t \geqslant 0$, $g(t) \in \varphi^{-1}(\mathbb{D}_{r'})$.

Note that $\varphi^{-1}(\mathbb{D}_{r'})$ is bounded, open and connected as the image of a bounded, open and (arc-)connected subset by a (non-constant) holomorphic function. So let us prove that $\varphi^{-1}(\mathcal{H}_{r'})$ contains a Nevanlinna-Sokal disk by showing that there exists $\delta > 0$ such that $\varphi(\mathbb{S}_\delta) \subset \mathcal{H}_{r'}$. We proceed in two steps. We first prove that if $z \in \mathbb{S}_\delta$ then $\operatorname{Re}\varphi(z) \geqslant 0$. Indeed, there exists a holomorphic function $\chi$ on $V$ such that $\varphi(z) = -\beta_2 z + z^2 \chi(z)$. Let $\theta$ be an argument of $z$.

$$\operatorname{Re}\varphi(z) \geqslant 0 \iff |\beta_2||z|\cos(\theta) \geqslant |z|^2 \cos(2\theta)\operatorname{Re}\chi(z) - |z|^2 \sin(2\theta)\operatorname{Im}\chi(z).$$

If $|z| = 0$ then $\operatorname{Re}\varphi(z) = 0$. Otherwise,

$$\operatorname{Re}\varphi(z) \geqslant 0 \iff |\beta_2|\cos(\theta) \geqslant |z|\cos(2\theta)\operatorname{Re}\chi(z) - |z|\sin(2\theta)\operatorname{Im}\chi(z).$$

But

$$|z|\cos(2\theta)\operatorname{Re}\chi(z) - |z|\sin(2\theta)\operatorname{Im}\chi(z) \leqslant 2\delta K_\delta \cos(\theta)\big(|\cos(2\theta)| + |\sin(2\theta)|\big)$$
$$\leqslant 4\delta K_\delta \cos(\theta),$$

where $K_\delta = \sup_{z \in \overline{\mathbb{S}}_\delta} |\chi(z)|$ and $\lim_{\delta \to 0} K_\delta = 0$. Thus, for $\delta$ small enough, $4\delta K_\delta < |\beta_2|$ and $\operatorname{Re}\varphi(z) \geqslant 0$.

We then show that $|\varphi(z)| \leqslant \frac{r'}{1 + 3\pi\frac{|\beta_3|}{\beta_2^2}r'}$:

$$|\varphi(z)| \leqslant |\beta_2||z| + K_\delta |z|^2 \leqslant 2|\beta_2|\delta\left(1 + 2\frac{\delta K_\delta}{|\beta_2|}\right) \leqslant 3|\beta_2|\delta.$$

As a consequence, fixing

$$3|\beta_2|\delta = \frac{r'}{1 + 3\pi\frac{|\beta_3|}{\beta_2^2}r'}, \quad \epsilon = \frac{2}{|\beta_2|}r',$$

the theorem is proved. $\quad\square$

If the initial value $g_r$ is real and $h$ real-valued, we can be more precise:

**Theorem 6.5** *There exists $g_c \in \mathbb{R}_+^*$ such that for all $g_r$ real in $(0, g_c)$, the Cauchy problem eq. (6.15) has a unique decreasing solution $g$ defined on $\mathbb{R}_+$. Moreover let $\epsilon$ be a positive real number smaller than 1. Then there exists a positive real number $\alpha(\epsilon)$ (smaller than 1) such that if $g_r \in (0, \alpha g_c)$, $g$ satisfies*

$$\frac{g_r}{1 - \beta_2 g_r t + \frac{\beta_3^-}{\beta_2} g_r \log(1 - \beta_2 g_r t) + \frac{\beta_3^-}{\beta_2} g_r \phi_-(t)} < g(t)$$
$$< \frac{g_r}{1 - \beta_2 g_r t + \frac{\beta_3^+}{\beta_2} g_r \log(1 - \beta_2 g_r t) + \frac{\beta_3^+}{\beta_2} g_r \phi_+(t)}, \quad (6.17)$$

*with $\beta_3^- := (1 - \operatorname{sgn}(\beta_3)\epsilon)\beta_3$, $\beta_3^+ := (1 + \operatorname{sgn}(\beta_3)\epsilon)\beta_3$ and $\phi_-, \phi_+$ two bounded functions on $\mathbb{R}_+$.*

**Proof.** — By the Cauchy-Lipschitz theorem, for all $g_r \in U \cap \mathbb{R}$, there exists a unique solution of the Cauchy problem eq. (6.15) defined on $[0, T)$ for some $T > 0$. Let $a : U \cap \mathbb{R} \to \mathbb{R}$ be defined as

$$f(t, x) := \beta_2 x^2 a(x),$$

so that $a(x) = 1 + \beta_{3,2} x + \frac{1}{\beta_2} x^2 h(x)$. Let $g_c$ be the smallest positive zero of $a$ in $U \cap \mathbb{R}$ if it exists and $\sup U \cap \mathbb{R}$ otherwise. As $f(t, x)$ is negative if $x \in (0, g_c)$, by unicity of the solutions of the Cauchy problem eq. (6.15), for all $t \in [0, T)$, $f(t, g(t))$ is negative. As a consequence, $0 < g(t) < g_r$ and $g$ is in fact defined on $\mathbb{R}_+$ (and decreasing).

Moreover if $g_r$ is small enough (say smaller than $\alpha g_c$),

$$\beta_2 g^2 + (1 - \text{sgn}(\beta_3)\epsilon)\beta_3 g^3 < \beta_2 g^2 + \beta_3 g^3 + g^4 h(g) < \beta_2 g^2 + (1 + \text{sgn}(\beta_3)\epsilon)\beta_3 g^3$$

for all $t \geqslant 0$ and by Theorem 6.3, $g$ satisfies eq. (6.17). $\square$

# Acknowledgments

F. V.-T. thanks É. Fouassier and T. Lepoutre for their kind explanations about the Cauchy-Lipschitz theorem and other basic facts about ODEs. He also thanks N.-V. Dang for many interesting and useful discussions and for advising him to contact F. Loray. It is a pleasure for us to warmly thank F. Loray who kindly explained us basic (but nevertheless very useful) aspects of holomorphic dynamics. V. R. deeply thanks his coauthor, and all those who encourage him, at a critical moment, to resume his scientific activity.

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
