# Peer review of "Can we make sense out of "Tensor Field Theory"?"

_SciPost Physics Core, doi:SciPost Phys. Core 4, 029 (2021)_

## Round 1 · Referee Report · Anonymous (Referee 1) · 2021-6-13

Strengths

1) Opens a new pathway in an existing or a new research direction, with clear potential for multipronged follow-up work: The T_5^4 model seems like a particularly interesting model to understand as it is asymptotically free and just renormalizable and in many ways seems far more accessible than non-Abelian gauge theories - the paper carries out several important first steps in rigorously constructing this model in the context of constructive field theory

2) Provides a novel and synergetic link between different research areas - This investigation of the T_5^4 model is likely accessible to a much wider audience than the random tensor community, such as the wider constructive field theory community and also perhaps the stochastic analysis community

Weaknesses

1) As the paper progresses, it becomes much less accessible to readers unfamiliar with [Riv91], random tensor models, and intermediate field methods

Report

I believe the article under review meets the journal's acceptance criteria and I strong recommend it for acceptance with minor revisions. The paper begins the analysis of a very interesting model and presents its features in a way that is more accessible than most papers on similar topics. The technical level is good and it is carefully written.

In addition to a few minor typos listed below, I also make some suggestions about changes that I believe would help make the paper easier to follow for those in other areas (such as stochastic quantization). While making the paper self-contained is not a reasonable aim, some additional material on what it means to construct a model like this, ribbon graphs, and/or intermediate field maps would help make things more accessible to a wider audience but aren't essential for the paper being published.

Requested changes

Minor points 1) Second paragraph of Introduction - "electroweek" should be "electroweak" 2) Section 1.3, "To explicit the interaction (...).." should be "To make the interaction (...) explicit..." 3) The formula for the interaction at the bottom of page 5 should probably come with a sentence indicating what the sum over bold indices is 4) Pg 7, "c-deges" should be "c-edges"

Areas where some detail/exposition might makes things much easier for people in areas 1) It might be good, perhaps in the introduction, to describe a bit what the steps of a constructive program for this model could be 2) Towards the beginning of section 2, a computation showing how one generates stranded graphs in perturbation theory 3) Section 2.3 could give a bit more detail on the graphical expansion with intermediate fields

---

## Round 2 · Author Response

We would like to thank the referee for her/his suggestions in order to improve and clarify the content of our manuscript. We also thank her/him for pointing out typos.

---

## Round 2 · List of Changes

Apart from correcting the typos indicated by the referee, we added two paragraphs at the end of the introduction (see p. 4) to precise what means the constructive program about the T^4_5 model and what are its first next steps.
The referee also suggested we add some explanations about how tensor models generate tensor graphs and about the graphical expansion in the intermediate field representation. This in fact is nothing but the very standard procedure of deriving Feynman graph expansions from a quantum field theory action. Moreover we think that our paper cannot easily be made accessible to readers who do not know enough QFT. Even if we would have added explanations about the different Feynman graph expansions used in the manuscript, readers who do not know QFT enough would have been unable to understand the renormalization parts (Sections 3 and 4). This is why we did not add any other explanations about tensor and intermediate field Feynman graphs.
The referee also suggested we add some explanations about how tensor models generate tensor graphs and about the graphical expansion in the intermediate field representation. This in fact is nothing but the very standard procedure of deriving Feynman graph expansions from a quantum field theory action. Moreover we think that our paper cannot easily be made accessible to readers who do not know enough QFT. Even if we would have added explanations about the different Feynman graph expansions used in the manuscript, readers who do not know QFT enough would have been unable to understand the renormalization parts (Sections 3 and 4). This is why we did not add any other explanations about tensor and intermediate field Feynman graphs.

---

## Editorial Decision

published